# On Preference Optimization in Large Language Models Under Pure Semantic Preferences

## Abstract

Large language models (LLMs) are typically aligned with human preferences through methods such as direct preference optimization (DPO). While empirically successful, these approaches face well-known limitations, including length bias, reward hacking, binary preference assumptions, and the aggregation of heterogeneous preferences into a single scalar signal. In this work, we take an inverse perspective: rather than attempting to resolve these issues directly, we investigate an idealized setting, which we call the *pure semantic preference scenario*, where such confounding factors are absent. To formalize this setting, we decompose the log-likelihood preference gap between two semantically equivalent generations into three additive components: a length alignment gap, a syntactic alignment gap, and a semantic alignment gap, and study the regime in which the length and syntactic gaps are controlled to be zero, so that observed preferences reflect semantics alone. We show that even in this idealized setting, existing alignment methods still do not fully capture the preference. Our analysis further reveals that (i) on-policy algorithms align more effectively, (ii) models trained without an explicit reference model perform better, and (iii) preference-model-based approaches consistently outperform reward-model-based approaches. Finally, motivated by these observations, we propose a lightweight preference-matching optimization (PMO) with a closed-form optimum that is well-suited to the pure semantic setting. Experiments on both practical and idealized settings demonstrate performance comparable to standard alignment baselines in the practical setting, while yielding clearer theoretical interpretation and improved results in the pure semantic setting.

## 1 Introduction

Large language models (LLMs) such as GPT-5 and Claude Sonnet-4 have demonstrated impressive performance across a wide range of tasks, including program synthesis, quantitative analysis, basic mathematics, and reasoning abilities (Hurst et al., 2024; Anthropic, 2024; Chowdhery et al., 2023; Touvron et al., 2023; Ji et al., 2025). Their rapid progress has led to deployment in decision-making contexts that, until recently, were thought to require exclusively human judgment (Bubeck et al., 2023; Eloundou et al., 2024).

One of a key factor behind this success is alignment: the ability of LLMs to adapt their outputs to human expectations, values, and conversational norms. The most widely adopted techniques for this purpose are reinforcement learning from human feedback (RLHF) (Ouyang et al., 2022; Casper et al.; Dong et al., 2024) and direct preference optimization (DPO) (Rafailov et al., 2023). RLHF proceeds in two stages. First, a reward model is trained on human preference data, often using the Bradley–Terry–Luce (BTL) model to transform pairwise judgments into a latent scoring function (Bradley & Terry, 1952; Luce, 2012). A higher reward assigned to a candidate response indicates that labelers favor it over alternatives, and this is taken as a proxy for broader human preference. Next, the base LLM is fine-tuned against this reward model, steering it toward producing responses with high predicted preference scores.

Despite their empirical success, preference alignment methods such as RLHF and DPO face a number of fundamental limitations. One major issue is length bias: models tend to favor longer responses that increase the probability of satisfying surface-level heuristics, even when verbosity harms clarity or faithfulness. Closely

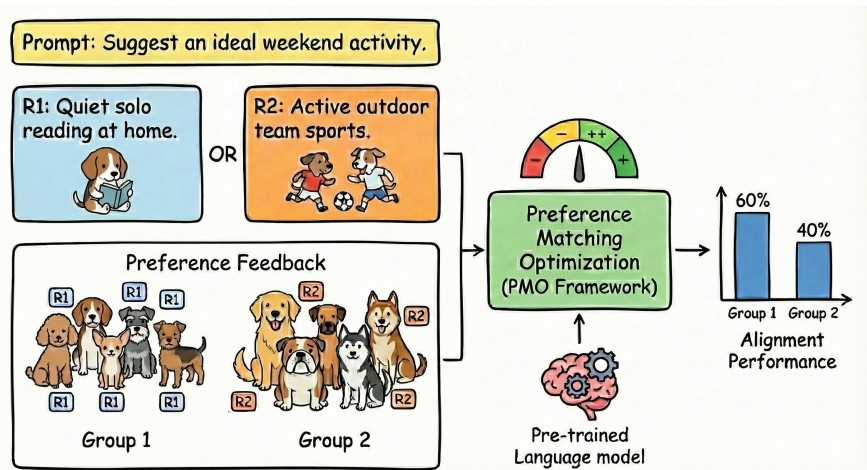

Figure 1: An illustrative example of pure semantic preference scenario, constructed using (i) minimal pairs, (ii) semantic difference, and (iii) probabilistic preferences.

related is reward hacking, where models exploit spurious correlations in the reward model or feedback process, producing outputs that optimize proxy signals while drifting from genuine human intent. A further limitation lies in the binary preference assumption: many frameworks reduce rich human judgments to a simplistic "winner" versus "loser" comparison, neglecting the subtleties of neutrality, partial agreement, or multi-dimensional trade-offs. This is compounded by the aggregation problem, where diverse annotator preferences are collapsed into a single scalar reward, often masking minority viewpoints and reinforcing majority bias.

It is commonly recognized that the main barriers to alignment include, but are not limited to, the challenges outlined above. Consequently, numerous variants of RLHF and DPO have been proposed; see Section 2 for further details. Motivated by these challenges, we decompose alignment gap into three components: length alignment gap, syntactic alignment gap, and semantic alignment gap. Under this view, reward hacking can be interpreted as an excessive emphasis on the first two components while insufficiently capturing the third.

*How do preference alignment methods perform under a purely semantic preference setting?*

The pure semantic preference scenario for preference alignment algorithms is constructed as follows. *Minimal pairs:* for any given prompt, the two candidate responses are of identical length, ensuring that preferences are not influenced by response length. The two responses share the same syntactic structure, so that preferences are not affected by stylistic or structural variations. Under these two conditions, the responses differ only in a single word (or phrase), which represents the main meaning of the sentence. *Semantic difference:* there is no notion of truth or falsity between the two responses. *Probabilistic Preference:* there is no strict binary preference; instead, there exists a probability $p \in [0, 1]$ such that $p$ fraction of people prefer the first response while $1 - p$ fraction prefer the second. An illustrative example is provided in Figure 1.

In other words, under this setting, all alignment approaches exhibit zero length and syntactic bias. Their performance on pure semantic bias therefore offers a more direct view of reward hacking.

Next, we evaluate the performance of various alignment methods on the pure semantic preference scenario using models. We find that in this idealized setting, where responses do not differ in length or sentence pattern, most alignment methods still do not fully capture the preference. We observe a pronounced preference–accuracy trade-off: improving alignment with diverse human preferences inevitably reduces accuracy, while prioritizing accuracy diminishes alignment with those preferences. In addition, within these methods, our findings can be summarized as follows: (i) on-policy algorithms align more effectively with pure semantic preferences; (ii) models trained without an explicit reference model perform better; and (iii) preference-model–based approaches (e.g., NLHF) consistently outperform reward-model–based approaches (e.g., RLHF).

In this paper, we use the terms **reward-model–based** and **preference-model–based** to distinguish two broad families of alignment pipelines. By reward-model–based we mean RLHF-style, typically on-policy methods that optimize a policy against a learned scalar reward model via reinforcement learning updates (e.g.,

PPO, NashMD). By preference-model–based we mean objectives that operate directly on pairwise preference data by matching probability ratios or scores without training an explicit reward model (e.g., DPO, CDPO, IPO, SimPO, CPO, PMO). We adopt this terminology purely for expository clarity; Section 2 maps these categories back to the standard RLHF and DPO formulations used in prior work.

In our experiment, the observed preference–accuracy trade-off arises from the reliance on a reference model and seems inevitable. To probe this, we first analyze a reference-free objective: its optimum recovers the ground-truth Bradley–Terry probabilities, exactly matching the target preference distribution. We further note that dropping the reference term in DPO is analogous to replacing the KL control in RLHF with an entropy regularizer, yielding a maximum-entropy formulation that curbs overconfident collapse and better preserves probabilistic preferences. Motivated by this, we adopt an RL objective that combines entropy and KL regularization, jointly preserving probabilistic preferences while maintaining accuracy—achieving a better trade-off. This objective is equivalent to Soft Preference Learning (SPL; (Slocum et al., 2025)) up to a simple reparameterization; we build on this objective but study it through our controlled semantic probability-matching lens and the resulting reference-attenuation interpretation.

Finally, we return to the practical setting by fine-tuning on the UltraFeedback dataset and evaluating performance across five benchmark tasks. In these experiments, we find that preference matching optimization attains performance comparable to existing methods.

## 2 Related Work

**Alignment with human preference.** DPO reframes RLHF as supervised ratio matching, improving stability and sample efficiency, but its implicit KL can compress diversity and bias toward majority styles, limiting peak accuracy without careful regularization (Rafailov et al., 2023). cDPO calibrates/conditions preference learning to correct annotator noise/context bias, recovering win rates while reducing shifts on near-tie pairs to preserve minority or user-specific preferences (Mitchell, 2024). IPO relaxes Bradley–Terry assumptions by matching scores directly, improving robustness under misspecification and heterogeneous feedback to preserve calibration and minority preferences with competitive accuracy (Azar et al., 2024). SimPO removes the fixed reference and uses a margin-based objective that often boosts win rate/accuracy, but risks drift unless margins and entropy are adaptively controlled (Meng et al., 2024). CPO replaces KL with chi-squared divergence, enabling larger yet controlled steps and improving the accuracy–preference Pareto frontier by avoiding KL's asymmetric pressure (Xu et al., 2024). PPO-based RLHF can raise reward and accuracy via exploration but is prone to over-optimization, instability, and diversity loss due to KL pressure and reward-model coupling (Schulman et al., 2017). Nash-MD frames alignment as a mixed-strategy equilibrium; mirror-descent updates and mixture sampling act as an implicit trust region to improve accuracy while maintaining pluralistic preferences (Munos et al., 2024). H-DPO adds entropy control by scaling the reverse-KL entropy term, yielding sharper, more mode-seeking policies that improve accuracy and pass@k without post-hoc temperature tuning (Omura et al., 2024). Soft Preference Learning (SPL; (Slocum et al., 2025)) decouples entropy and cross-entropy regularization.

**Diversity in human preferences.** Most alignment methods average annotator preferences, overlooking diversity rooted in social and cultural backgrounds; key drivers include socio-demographics, personal bias and context subjectivity, imperfect preferences, and linguistic ambiguity or missing context (Casper et al.; Kaufmann et al., 2023; Chakraborty et al., 2024; Aroyo et al., 2023; Denton et al., 2021; Vogels, 2021; Sandri et al., 2023).

## 3 Pure Semantic Preference on Synthetic Dataset

In this section, we first study the following question:

*How to extract semantic preference from general preference?*

Given an input $x$ and two alternative responses $y_1$ and $y_2$ with length $n_1$ and $n_2$, we measure the model's *preference* between them by a log-likelihood difference, $G(y_1, y_2) = \log P_\theta(y_1 \mid x) - \log P_\theta(y_2 \mid x)$, which we

call an *alignment gap.* To attribute this gap to interpretable factors, we introduce two intermediate responses: (i) a *length-aligned* composition $y_2''$ obtained from $y_2$ that matches the length of $y_1$ while preserving meaning, and (ii) a *syntax-aligned* composition $y_2'$ that preserves the meaning of $y_2''$ but matches the syntactic form of $y_1$.

**Notation.** Let $m : \mathcal{V}^* \to \mathcal{M}$ be a semantic representation map , where $\mathcal{M}$ is a meaning set, $\mathcal{V}$ be a vocabulary and $\mathcal{V}^* = \bigcup_{n \geq 1} \mathcal{V}^n$; define semantic equivalence by $y \sim_{\text{sem}} y'$ iff $m(y) = m(y')$. For any meaning $M \in \mathcal{M}$, let the equivalence class be $\mathcal{Y}(M) := \{y \in \mathcal{V}^* : m(y) = M\}$. We consider two responses $y_1 = [T_1^1, \ldots, T_{n_1}^1] \in \mathcal{V}^{n_1}$ and $y_2 = [T_1^2, \ldots, T_{n_2}^2] \in \mathcal{V}^{n_2}$ with $n_2 > n_1$, where $\mathcal{V}^{n_j} = \bigcup_{n_j} \mathcal{V}$; $T_i^j$, $i = 1, \ldots, n_j$, are different tokens of $y_j$ for $j = 1, 2$.

To formally address the potential entanglement between length, syntax, and semantics, as these factors are often naturally correlated in human language, we introduce the following working hypothesis to justify the identifiability of our decomposition. The semantic alignment gap is the residual difference once length and syntax are controlled. This yields an exact decomposition of the overall gap into length, syntactic, and semantic components.

**Working Hypothesis 3.1 (Orthogonal Attribution of Alignment)** *We assume that the total alignment gap $G(y_1, y_2) = \log P_\theta(y_1 \mid x) - \log P_\theta(y_2 \mid x)$ can be decomposed into a sequence of additive marginal contributions by traversing a path of intermediate latent representations:*

1. *There exists an intermediate response $y_2''$ that shares the same meaning as $y_2$ but matches the length of $y_1$. The **length gap** is defined as $\log P_\theta(y_2'' \mid x) - \log P_\theta(y_2 \mid x)$.*

2. *There exists an intermediate response $y_2'$ that shares the same meaning as $y_2''$ but matches both the length and syntactic structure of $y_1$. The **syntax gap**, although naturally correlated with length, is defined as the residual gap $\log P_\theta(y_2' \mid x) - \log P_\theta(y_2'' \mid x)$ conditioned on the length alignment.*

3. *The **semantic gap** is defined as the residual gap $\log P_\theta(y_1 \mid x) - \log P_\theta(y_2' \mid x)$ conditioned on both the length and syntactic alignment.*

To further construct such a decomposition, we assume that the semantic difference between $y_1$ and $y_2$ is captured by two anchor tokens $T_{i_1}^1$ and $T_{i_2}^2$ (or, more generally, by short token sequences), where $i_1$ and $i_2$ are the positions of these anchor tokens.

**Length-aligned composition.** To isolate the effect of length, we introduce an intermediate response $y_2''$ that preserves the meaning of the longer realization $y_2$ while matching the target length $n_1$. We construct $y_2''$ by selecting $n_1$ token positions from $y_2$ in their original order (i.e., a subsequence) and additionally require the designated anchor tokens $T_{i_2}^2$ are retained to avoid dropping salient content. Formally, we define the length-aligned response

$$y_2'' := \Pi_I(y_2) = [T_1'', \ldots, T_{n_1}''], \tag{1}$$

where, $T_i''$ are the tokens of $y_2''$, $i = 1, \ldots, n_1$, such that $y_2''$ is a subsequence of $y_2$ that contains $T_k^2$ and $m(y_2'') = m(y_2) = M$. Then, the log-likelihood difference between $y_2''$ and $y_2$ can be interpreted as a length preference under fixed meaning.

**Syntax-aligned composition.** Since the core semantic content of $y_2$ is encapsulated in the anchor token $T_{i_2}^2$, we treat $y_1$ as a syntactic template. We then construct the syntax-aligned response $y_2'$ by substituting the anchor token of $y_1$ with that of $y_2$. Formally, let $y_1 = [T_1^1, \ldots, T_{i_1}^1, \ldots, T_{n_1}^1]$. We define:

$$y_2' := [T_1^1, \ldots, T_{i_2}^2, \ldots, T_{n_1}^1], \tag{2}$$

under the assumption that $y_2'$ remains grammatically correct. By this construction, $y_2'$ inherits the length and syntactic skeleton of $y_1$ while adopting the key semantic component of $y_2$.

**A decomposition of Alignment Gaps.** For any choice of $(y_2'', y_2')$ satisfying Equation (1)–Equation (2), the following identity holds:

$$\log P_\theta(y_1 \mid x) - \log P_\theta(y_2 \mid x)$$
$$= \underbrace{\left[ \log P_\theta(y_1 \mid x) - \log P_\theta(y_2' \mid x) \right]}_{\Delta_{\text{sem}}} + \underbrace{\left[ \log P_\theta(y_2' \mid x) - \log P_\theta(y_2'' \mid x) \right]}_{\Delta_{\text{syn}}} + \underbrace{\left[ \log P_\theta(y_2'' \mid x) - \log P_\theta(y_2 \mid x) \right]}_{\Delta_{\text{len}}} . \quad (3)$$

We call the first term $\Delta_{sem}$ semantic alignment gap, $\Delta_{syn}$ syntax alignment gap, and $\Delta_{len}$ length alignment gap.

**Discussion on Path-Dependence.** Under this hypothesis, the decomposition in Equation (3) effectively treats the alignment gap as a path-dependent line integral in the feature space. By constructing $y_2'$ and $y_2''$ as sequential "projections" of $y_2$ onto the constraints of $y_1$, we ensure that $\Delta_{\text{sem}}$ captures the *residual preference* that cannot be explained by surface-level features (length and syntax). This approach follows the *ceteris paribus* principle, allowing for the isolation of specific model biases even when underlying factors are statistically intertwined.

However, finding a practical and reliable algorithm to construct the intermediate pair $(y_2', y_2'')$—while strictly enforcing semantic equivalence and isolating syntax and length—can be nontrivial. Rather than explicitly solving for $(y_2', y_2'')$, we curate a dataset in which candidate pairs are controlled by construction so that length and syntactic form are matched. Concretely, we design the data such that the two compared responses have identical length and (up to a chosen representation) identical syntactic structure, which implies $\Delta_{\text{len}} = 0$ and $\Delta_{\text{syn}} = 0$ by definition of the factorized gaps. Under these controls, the overall alignment gap reduces to the semantic alignment gap, enabling us to study and optimize semantic alignment without requiring an explicit intermediate-response generation algorithm. Then, we study the next question:

*How do preference alignment methods align semantic preference?*

**Pure semantic preference scenario.** To isolate alignment on meaning rather than form, we first instantiate a controlled "pure semantic" scenario in which all non-semantic confounds are neutralized. For any given prompt, we construct two candidate responses that are (i) identical in length, eliminating length-induced preferences and token-count biases, and (ii) matched in sentence pattern, sharing the same syntactic template and differing only by a single lexical item occupying the same position—the main content noun (e.g., "I favor tea" vs. "I favor coffee"). By design, neither candidate is more or less "true": the contrast is semantically neutral with respect to factuality, so correctness cannot explain preferences. Instead of a hard choice, we posit a probabilistic preference: there exists a target probability $p \in [0,1]$ that the first response is preferred, with $1 - p$ for the second. Under these constraints, any difference in model behavior can be attributed to the intended semantic substitution, and alignment reduces to matching the target pairwise preference probability $p$ in a setting free from length, format, or stylistic confounds.

We introduce a preference dataset tailored to isolate semantic choices while removing confounds such as length, formatting, or discourse structure. Each instance is a minimal pair (Warstadt et al., 2020) built from a single prompt and two completions that differ only by one lexical item in the same position (e.g., "cola" vs. "pepsi", "tea" vs. "coffee"). Unlike conventional binary preference data used in RLHF (Rafailov et al., 2023; Li et al., 2023), we attach a soft target—the probability that one completion is preferred over the other—explicitly provided in the dataset.

**Design principles.** (i) Minimal pairs. For a fixed prompt $x$, we create two responses $y_A, y_B$ by substituting a single content word at a pre-specified slot in a templated response, ensuring that surface form, punctuation, and syntax are otherwise identical. This minimal-pair design targets "pure" semantic variation (Warstadt et al., 2020). (ii) Soft preferences. We annotate each pair $(x, y_A, y_B)$ with a target probability $p \in (0, 1)$ that $y_A$ is preferred over $y_B$; the complementary probability for $y_B$ is $1 - p$. The value $p$ is synthetically specified (fixed RNG seed for reproducibility) because the pure semantic setting lacks scalable, naturally available preference-frequency targets for subjective minimal pairs; the resulting supervision is a low-stakes probe of

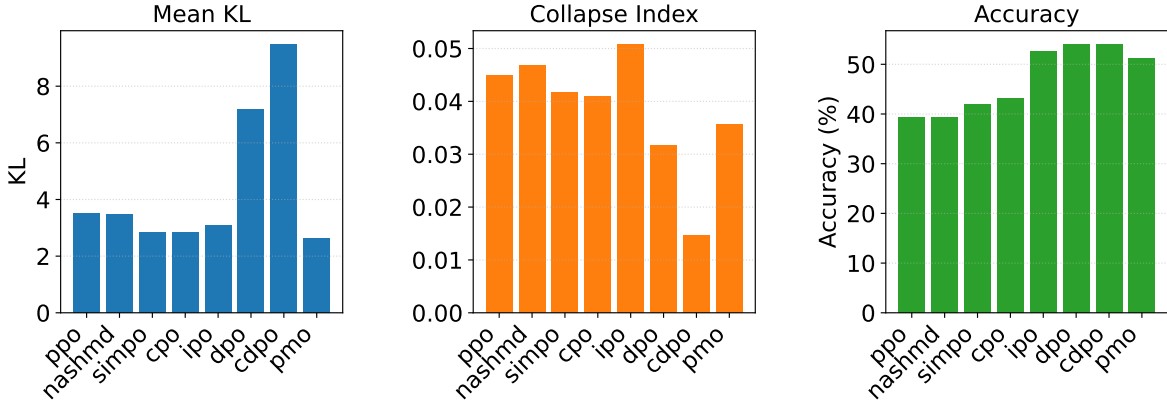

Figure 2: Preference–accuracy trade-off on the Llama model, measured by Mean KL, collapse index (PCI), and accuracy (%), for PPO, NashMD, SimPO, CPO, IPO, DPO, CDPO, and PMO.

probability matching/collapse rather than a claim about semantic "truth". (iii) Dialogue form. Instances are packaged as short user–assistant turns to mirror RLHF preference data schemas (Rafailov et al., 2023; Li et al., 2023) while preserving strict control over the single-word contrast [1].

We analyze three metrics reported for eight preference-learning methods across three backbones (Qwen, Gemma, Llama): (i) Accuracy, (ii) Mean KL, interpreted as the KL divergence between the target label distribution and the model's predicted distribution (lower is better), and (iii) the Preference Collapse Index (PCI), defined as

$$\text{PCI} = \frac{1}{n} \sum_{i=1}^{n} \min\{p_i,\, 1 - p_i\},$$

where $p_i \in [0, 1]$ is the predicted probability of the positive class on example $i$. PCI measures the average distance of predictions from deterministic extremes: lower PCI indicates stronger collapse toward a single option, whereas higher PCI indicates more uncertainty (with PCI = 0.5 achieved at $p_i = 0.5$ for all $i$). In light of prior observations that preference collapse can undermine the faithful representation of distributional preferences, suppressing minority outcomes, we interpret very low PCI as a warning signal of overconfident, potentially collapsed behavior, especially when accompanied by large KL.

**Tradeoff between preference and accuracy.** Figure 2 reports results on our synthetic Llama setup across strong preference-optimization baselines (DPO, CDPO, IPO, SimPO, CPO, PPO, NashMD). We quantify preference preservation with Mean KL (lower is better) and the PCI (higher is better). Both metrics consistently indicate that reference-free objectives (e.g., SimPO, CPO) align more faithfully with the target probabilities, exhibiting lower KL and higher PCI. However, when accuracy is considered, a clear tradeoff emerges: methods like DPO and CDPO that push predictions toward decisive extremes can improve accuracy but typically inflate KL and reduce PCI (i.e., more collapse), whereas methods that maintain calibrated distributions improve KL/PCI but may concede some accuracy.

## 4 Preference Matching Optimization

### 4.1 Preliminaries

**RLHF.** Let $\pi_\phi(y|x)$ be the probability distribution of the responses given a prompt $x$, where $\phi$ denotes the weights of the LLM. The goal of RLHF is to maximize the expected reward with a KL penalty between the RLHF model and the reference model. The loss function of is

$$\max_\phi \mathbb{E}_{x \sim \rho} \mathbb{E}_{y \sim \pi_\phi(\cdot|x)} r(x, y) - \beta D_{\text{KL}}(\pi_\phi(y|x) \| \pi_{\text{ref}}(y|x)), \tag{4}$$

---

[1]We leave the details of our dataset in the Appendix A.2.

where $\beta > 0$ is a parameter controlling the deviation from the base reference policy $\pi_{\text{ref}}$.

**DPO.** The DPO method (Rafailov et al., 2023) is to directly optimize of the policy without explicitly training the reward function in a supervised manner:

$$-\mathbb{E}_{(x,y_w,y_l)} \log \sigma \left( \beta \log \frac{\pi_\phi(y_w|x)}{\pi_{\text{ref}}(y_w|x)} - \beta \log \frac{\pi_\phi(y_l|x)}{\pi_{\text{ref}}(y_l|x)} \right).$$

**SimPO.** The objective of SimPO (Meng et al., 2024) can be written as

$$-\mathbb{E}_{(x,y_w,y_l)} \log \sigma \left( \frac{\beta}{|y_w|} \log \pi_\phi(y_w|x) - \frac{\beta}{|y_l|} \log \pi_\phi(y_l|x) - \gamma \right), \tag{5}$$

where $|y|$ denotes the length of a response[2], and $\gamma$ is the reward margin, with the preference probability expressed as $p(y_w \succ y_l|x) = \sigma(r(x, y_w) - r(x, y_l) - \gamma)$.

## 4.2 Mathematical Formulations

The tradeoff arises from the reliance on a reference model and seems inevitable. But is that truly the case? To investigate, we first examine a reference-free objective. We consider SimPO as an illustrative example, with the corresponding analysis for other compared algorithms deferred to Appendix B. The following proposition provides its corresponding RLHF objective and optimal policy.

**Proposition 4.1** *Let $\beta' = \beta/|y|$, and let $r_\gamma(x, y)$ denote a reward model with a reward margin $\gamma$. Then minimizing the direct alignment objective in Equation 5 is equivalent to solving the reinforcement learning problem*

$$\max_\phi \ \mathbb{E}_{x\sim\rho} \mathbb{E}_{y\sim\pi_\phi(\cdot|x)} \Big[ r_\gamma(x, y) \Big] + \beta' H\big(\pi_\phi(\cdot|x)\big), \tag{6}$$

*whose optimal policy is given by $\pi^\star(y|x) = \exp\left( \frac{1}{\beta'} r_\gamma(x, y) \right) / \sum_{y'} \exp\left( \frac{1}{\beta'} r_\gamma(x, y') \right).$*

**Why do reference-free approaches better preserve probabilistic preference?** A direct consequence of Proposition 4.1 is that when $\beta = |y|$ and $\gamma = 0$, the optimal solution coincides with the ground-truth BT preference. In other words, SimPO can recover the target probabilistic preference with appropriately chosen parameters. By contrast, for reference-based approaches such as DPO, this is not possible. Recall that the optimal solution (Rafailov et al. (2023), cf. Equation (4)) of DPO is given by

$$\pi^\star(y|x) = \frac{\pi_{\text{ref}}(y|x) \exp(r(x, y)/\beta)}{\sum_{y'} \pi_{\text{ref}}(y'|x) \exp(r(x, y')/\beta)}.$$

Regardless of the choice of $\beta$, the influence of $\pi_{\text{ref}}$ cannot be removed, and thus the solution cannot exactly preserve the target probabilistic preference.

**Regularization.** A second observation from Proposition 4.1 is that removing the reference term in the DPO objective is equivalent to replacing the KL term in the RLHF objective with an entropy term. Maximizing entropy plays a key role in preserving the target preference. Notably, this perspective is not discussed from the original SimPO paper (Meng et al., 2024), where the reference model was removed primarily for computational and memory considerations.

Motivated by the observation that the ability of reference-free approaches to preserve probabilistic preferences arises primarily from the inclusion of the entropy term, rather than from the removal of the KL term, which in fact reduces accuracy, we consider the following RL problem that incorporates both the entropy and KL terms to achieve a better trade-off:

$$\max_\phi \mathbb{E}_{x\sim\rho} \mathbb{E}_{y\sim\pi_\phi(\cdot|x)} r(x, y) + \alpha H(\pi_\phi(y|x)) - \beta D_{\text{KL}}(\pi_\phi(y|x) \| \pi_{\text{ref}}(y|x)). \tag{7}$$

The following proposition provides its corresponding direct alignment objective and optimal policy.

---

[2]In the pure semantic preference scenario, $|y_w| = |y_l|$.

**Proposition 4.2** *Solving the reinforcement learning problem in Equation 7 is equivalent to the direct alignment objective*

$$-\mathbb{E}_{(x,y_w,y_l)} \log \sigma \left( (\alpha + \beta) \log \frac{\pi_\phi(y_w|x)}{\pi_{\mathrm{ref}}(y_w|x)^{\frac{\beta}{\alpha+\beta}}} - (\alpha + \beta) \log \frac{\pi_\phi(y_l|x)}{\pi_{\mathrm{ref}}(y_l|x)^{\frac{\beta}{\alpha+\beta}}} \right), \tag{8}$$

*whose optimal policy is* $\pi_\phi(y|x) = \frac{1}{Z(x)} \pi_{\mathrm{ref}}(y|x)^{\beta/(\alpha+\beta)} \exp\left( \frac{1}{\alpha+\beta} r(x,y) \right)$, *where the normalizing constant is* $Z(x) = \sum_y \pi_{\mathrm{ref}}(y|x)^{\beta/(\alpha+\beta)} \exp\left( \frac{1}{\alpha+\beta} r(x,y) \right)$.

Note that by substituting the KL divergence with the cross-entropy term in Equation 7, the objective recovers the SPL objective (Slocum et al., 2025). Specifically, the two objectives become identical under the reparameterization $\alpha \to \alpha - \beta$. In this sense, PMO is not presented as a fundamentally new objective function, but rather as a specialized formulation designed for the controlled semantic preference setting and the subsequent diagnostic analysis of probability matching versus collapse. Nevertheless, our framework emphasizes a parameter regime where the coefficient of the entropy term remains strictly positive relative to the KL term—a distinction not prioritized in the SPL literature. We retain the name PMO to highlight this specific focus and to maintain consistency with the reference-attenuation perspective established in Proposition 4.2.

**Relation to DPO.**   When $\alpha = 0$, PMO reduces to DPO. Moreover, if we set the value of $\alpha + \beta$ in PMO equal to the value of $\beta$ used in DPO, the two objectives differ only in the exponent on $\pi_{\mathrm{ref}}(y|x)$: PMO decreases this exponent from 1 to $\beta/(\alpha + \beta)$. This attenuation reduces the reference model's influence on the learned preference and thereby helps preserve the target probability distribution, while the KL regularizer maintains accuracy comparable to DPO.

**Relation to H-DPO.**   H-DPO reweights the reverse-KL by decomposing it into cross-entropy and entropy, effectively tuning the entropy term's contribution; the final loss is reward $+\ \alpha \cdot$ entropy - cross-entropy (App.B.2). PMO instead optimizes reward $+\ \alpha \cdot$ entropy - $\beta \cdot$ KL, which leads to a closed-form optimum whose policy multiplies the reference density by an exponent $\beta/(\alpha + \beta)$ (Prop.4.2). This explicit reference attenuation is central to preserving probabilistic preferences while retaining accuracy via a light anchor; it is not exposed in DPO (exponent 1) and is distinct from the cross-entropy view in H-DPO.

## 5   Experiment in Pure Semantic Preference Scenario

We evaluate off-policy preference-optimization baselines (CDPO, DPO, IPO, CPO, SimPO) and our off-policy PMO variants on our synthetic dataset, reporting per-task accuracy and macro-average. Besides, on-policy algorithms (PPO, NashMD) are also put into comparison. In PMO, $\alpha > 0$ scales (tempers) the preference scores that drive the update, and $\beta \geq 0$ controls the strength of the reference-model term; $\beta = 0$ denotes a reference-free objective.

### 5.1   Cross-Cutting Patterns and Implications

We observe a Pareto trade-off (Pareto frontier) among accuracy, PCI, and KL across backbones, consistent with multi-objective optimization behavior in RL (Liu et al., 2025) and recent evidence of metric trade-offs in RL-style training (e.g., accuracy vs consistency) (Park et al., 2025): (i) methods with the highest accuracy (DPO, and CDPO on Llama) systematically push PCI down (stronger collapse) and inflate KL (worse distance to ground truth); (ii) methods with the best KL (IPO, PMO, Nash-MD, PPO depending on backbone) maintain higher PCI (less collapse), reflecting better-calibrated probabilities that refrain from overconfident extremes; and (iii) intermediate methods (e.g., CPO, SimPO) trace the interior of this frontier.

These patterns are consistent with the interpretation of KL as a calibration or fit objective on the probability simplex: overconfident predictions (low PCI) penalize KL heavily when incorrect, whereas restrained probabilities (higher PCI) reduce KL by avoiding extreme errors. Simultaneously, pushing accuracy often benefits from confident decisions, which, when correct, boost accuracy despite degrading KL.

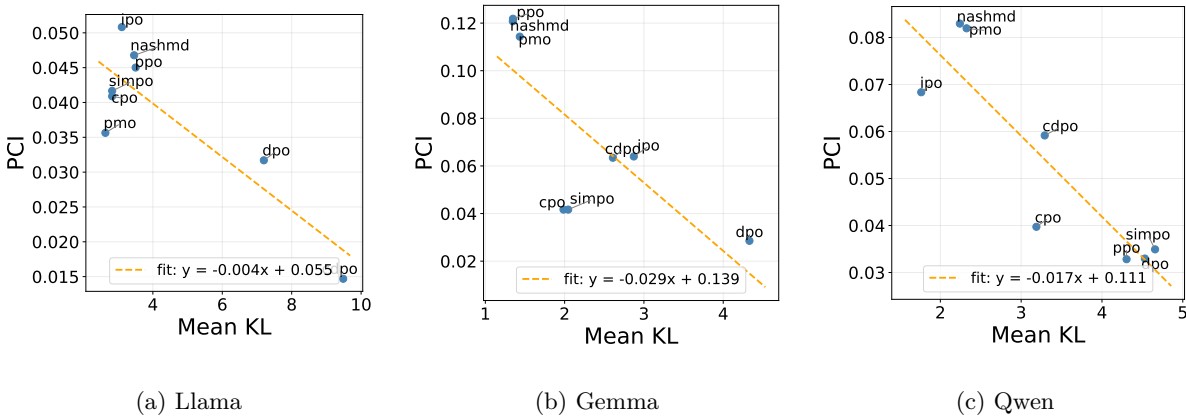

(a) Llama  (b) Gemma  (c) Qwen

Figure 3: Linear regressions between Mean KL and collapse index (PCI) for Llama, Gemma, and Qwen. Each point corresponds to one approach, and the dashed line indicates the least-squares fit.

## 5.2 Analysis of KL–PCI–Accuracy Trade-offs

**On-policy algorithms better align pure semantic preferences.** On Gemma, PPO/NashMD/PMO attain the lowest KL (1.35–1.43) and the highest PCI (0.114–0.122), whereas the most accurate off-policy method (DPO, 0.473) shows the worst KL (4.33) and strongest collapse (PCI 0.029), see Figure 3. On Llama, on-policy methods achieve favorable KL (3.459–3.503) and acceptable PCI (0.045–0.046), while off-policy CDPO/DPO maximize accuracy (0.541) at the expense of severe collapse (PCI 0.015–0.032) and very large KL (7.20–9.48). Qwen is mixed but consistent: on-policy PMO/NashMD have low KL (2.32/2.24) and the least collapse (PCI 0.082), with NashMD also reaching the second-best accuracy (0.446). These findings mirror broader evidence that on-policy RLHF tends to deliver better alignment than offline variants and that PPO-style training can outperform DPO given comparable data and settings.

**Models trained without an explicit reference model are better on collapse and KL.** Reference-free formulations (e.g., IPO, SimPO) avoid overconfident degeneration in two of the three backbones and often yield favorable KL–PCI trade-offs: IPO on Qwen achieves the lowest KL (1.76) with moderately high PCI (0.068), and on Llama achieves low KL (3.11) with the highest PCI (0.051). Although some reference-based, off-policy methods (e.g., DPO/CDPO) can peak in accuracy, this typically coincides with pronounced collapse and inflated KL. This aligns with reports that simpler, reference-free preference objectives like SimPO can match DPO performance while reducing complexity and sensitivity to hyperparameters (Meng et al., 2024).

**Trade-offs on 8B models.** Table 1 compares KL divergence, PCI, and downstream accuracy for different preference optimization methods on Llama3-8B and Qwen3-8B. On Llama3-8B, SimPO attains the lowest KL (1.73) while maintaining competitive accuracy, whereas CPO reaches the highest accuracy (0.53) at the cost of slightly larger KL and reduced PCI, and PMO yields the strongest PCI (0.13). On Qwen3-8B, PMO offers the best KL–PCI trade-off, achieving the lowest KL (1.66) and the highest PCI (0.16), while DPO delivers the best accuracy (0.50) with moderately higher KL. Overall, reference-free or partially reference-relaxed methods such as SimPO and PMO can match or closely approach the accuracy of DPO/CPO while often providing more favorable KL–PCI profiles, indicating improved robustness against representational collapse.

## 5.3 Ablation on Alpha and Beta in the Pure Semantic Setting

Table 2 evaluates how the score-scaling parameter $\alpha$ and the reference weight $\beta$ shape the trade-off among KL (probability alignment to the dataset targets; lower is better), PCI (anti-collapse; higher is better), and 0–1 accuracy on the pure semantic dataset where responses differ by a single content word and all non-semantic confounds are controlled. Three consistent patterns emerge. First, configurations that maximize accuracy (e.g., $\alpha$=0.05, $\beta$=0.05) do so by sharply degrading alignment: they yield the worst KL and the lowest PCI across all backbones (Gemma: KL 4.37, PCI 0.051, Acc 0.486; Llama: 5.46/0.033/0.556; Qwen: 8.21/0.030/0.556), indicating severe collapse and poor probability matching despite higher 0–1 accuracy. Second, moving to a stronger preference signal ($\alpha \approx 0.9$–1.0) while keeping a very light reference ($\beta \in \{0.05, 0.1\}$) substantially improves probabilistic fidelity and reduces collapse at a modest accuracy cost. For Gemma, $(\alpha, \beta)=(0.9, 0.1)$

Table 1: Mean KL, collapse index (PCI), and accuracy for Llama and Qwen 8b models.

| Backbone | Algorithm | KL ↓ | PCI ↑ | Accuracy ↑ |
|---|---|---|---|---|
| Llama3-8B | DPO | 2.1173 | 0.1284 | 0.4583 |
| | CPO | 1.8208 | 0.1160 | **0.5278** |
| | SimPO | **1.7322** | 0.1207 | 0.4722 |
| | PMO | 1.8749 | **0.1327** | 0.4722 |
| | IPO | 2.1659 | 0.1239 | 0.4444 |
| | CDPO | 2.1511 | 0.1282 | 0.4306 |
| Qwen3-8B | DPO | 2.1010 | 0.1247 | **0.5000** |
| | CPO | 2.1660 | 0.1226 | 0.4861 |
| | SimPO | 2.2535 | 0.1262 | 0.4167 |
| | PMO | **1.6633** | **0.1609** | 0.4583 |
| | IPO | 2.1707 | 0.1201 | 0.4722 |
| | CDPO | 2.3038 | 0.1024 | 0.4583 |

Table 2: Ablation study of $\alpha$ and $\beta$ on the synthetic dataset for Gemma, Llama, and Qwen, reporting KL, collapse index (PCI), and accuracy.

| alpha | beta | Model | KL | PCI | Accuracy |
|---|---|---|---|---|---|
| 0.05 | 0.05 | Gemma | 4.3658 | 0.0512 | **0.4861** |
| 0.5 | 0 | Gemma | 2.5124 | 0.1112 | 0.4167 |
| 0.9 | 0.1 | Gemma | **1.1632** | 0.1246 | 0.4167 |
| 0.95 | 0.05 | Gemma | 1.1933 | **0.1250** | 0.4306 |
| 1 | 0 | Gemma | 1.5178 | 0.1224 | 0.3889 |
| 0.05 | 0.05 | Llama | 5.4562 | 0.0332 | **0.5556** |
| 0.5 | 0 | Llama | 1.9248 | 0.1019 | 0.4028 |
| 0.9 | 0.1 | Llama | 1.9294 | 0.1025 | 0.4444 |
| 0.95 | 0.05 | Llama | **1.7057** | **0.1046** | 0.4444 |
| 1 | 0 | Llama | 2.1556 | 0.0912 | 0.4444 |
| 0.05 | 0.05 | Qwen | 8.2112 | 0.0297 | **0.5556** |
| 0.5 | 0 | Qwen | 3.7404 | 0.0781 | 0.4167 |
| 0.9 | 0.1 | Qwen | **1.5261** | 0.0984 | 0.4583 |
| 0.95 | 0.05 | Qwen | 1.6452 | **0.0991** | 0.4861 |
| 1 | 0 | Qwen | 2.3973 | 0.0803 | 0.4167 |

and $(0.95, 0.05)$ achieve the best alignment (KL 1.16–1.19; PCI 0.125), with accuracy 0.417–0.431; for Llama, $(0.95, 0.05)$ yields KL 1.71 and PCI 0.105 with accuracy 0.444; for Qwen, $(0.9, 0.1)$ reaches the best KL 1.53 with PCI 0.098 and accuracy 0.458, while $(0.95, 0.05)$ trades a small KL increase (1.65) for the highest accuracy in this block (0.486) with similarly high PCI (0.099). Third, reference-free training ($\beta{=}0$) at $\alpha \in \{0.5, 1.0\}$ underperforms the light-reference regime on alignment for Gemma/Llama and markedly so for Qwen (e.g., Qwen $\alpha{=}1, \beta{=}0$: KL 2.40, PCI 0.080, Acc 0.417), suggesting that a small reference term acts as a helpful calibration prior in this synthetic probability-matching task.

Backbone-wise, Gemma exhibits the strongest gains from adding a light reference at high $\alpha$ (KL drops from 2.51 at $\alpha{=}0.5, \beta{=}0$ to 1.16–1.19 at $\alpha{\approx}1, \beta{\in}\{0.05, 0.1\}$; PCI rises from 0.111 to $\approx$0.125), while Llama benefits similarly but with smaller absolute swings. Qwen shows a broad plateau near $(\alpha, \beta) \in \{(0.9, 0.1), (0.95, 0.05)\}$, both outperforming $\beta{=}0$ on KL and PCI and delivering competitive accuracy. Across all models, the settings that minimize KL also maximize PCI, reinforcing the earlier observation of a negative PCI–KL slope: better probability alignment coincides with less collapse. Practically, we recommend operating near $\alpha \in [0.9, 1.0]$ with a very light reference $\beta \in [0.05, 0.1]$ (Gemma: $(0.95, 0.05)$ or $(0.9, 0.1)$; Llama: $(0.95, 0.05)$; Qwen: $(0.9, 0.1)$ or $(0.95, 0.05)$). Extremely small $\alpha$ should be avoided despite its apparent accuracy gains, as it drives systematic miscalibration (high KL) and collapse (low PCI) in the pure semantic regime.

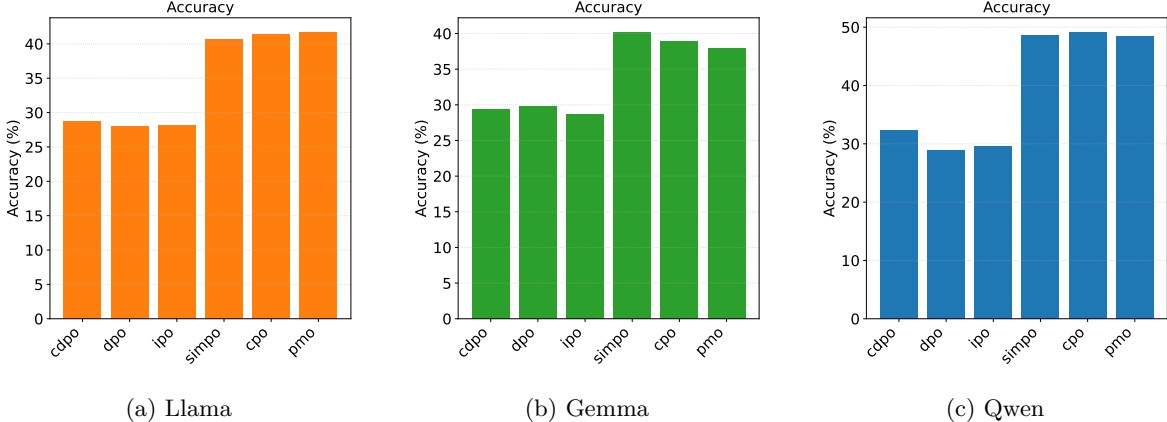

Figure 4: Average accuracy (%) on ARC-Challenge, HellaSwag, MMLU, TruthfulQA (MC1), and WinoGrande for CDPO, DPO, IPO, SimPO, CPO, and PMO on (a) Llama, (b) Gemma, and (c) Qwen.

**Practitioner note.** PCI is a diagnostic for overconfidence (collapse), not a target by itself. Tasks demanding decisive behavior can use lower $\alpha$ or higher $\beta$ to move toward the accuracy-seeking end of the frontier (see Table 1), whereas pluralistic or user-diverse settings may prefer higher $\alpha$ with a light reference $\beta \in [0.05, 0.1]$ (Table 1).

## 6 Benchmark and Ablation Analysis

We evaluate off-policy preference-optimization baselines (CDPO, DPO, IPO, CPO, SimPO) and our off-policy PMO variants on ARC-Challenge, HellaSwag, MMLU, TruthfulQA (MC1), and WinoGrande [3], reporting per-task accuracy and macro-average. In PMO, $\alpha > 0$ scales (tempers) the preference scores that drive the update, and $\beta \geq 0$ controls the strength of the reference-model term; $\beta = 0$ denotes a reference-free objective.

### 6.1 Overall Baseline Comparison

Without ablations, the strongest baselines are CPO/SimPO across backbones, see Figure 4. On Gemma-3B-1B, CPO/SimPO reach 0.389/0.402 average, substantially above CDPO/DPO/IPO (0.286–0.299) and PMO (0.293). On Qwen2.5-1.5B, CPO/SimPO achieve 0.492/0.486, clearly exceeding CDPO/DPO/IPO (0.288–0.323) and PMO (0.283). On Llama3-1B, CPO/SimPO obtain 0.415/0.407 versus 0.280–0.290 for CDPO/DPO/IPO/PMO. Gains are especially pronounced on HellaSwag and ARC, with strong improvements also on MMLU and WinoGrande.

### 6.2 Regularization and Reference Ablations: DPO vs. CPO vs. SimPO vs. PMO

Table 3 studies three knobs that often distinguish xPO objectives: (i) the reference term in DPO (here ablated by setting $\beta$=0), (ii) SimPO's length normalization and margin ($|y|, \gamma$), and (iii) the BC-style regularization in CPO (here denoted by $\lambda$). Conceptually, removing the reference collapses DPO toward a policy-only scoring; removing SimPO's length/margin reduces it to a policy-only Bradley–Terry loss; and turning off CPO's BC regularizer yields a pure preference objective. These manipulations are expected to make the objectives converge in behavior, consistent with analyses that relate SimPO to a length-normalized DPO family via mixing and show that length normalization and the margin term are the main sources of divergence across objectives (Meng et al., 2024; Azar et al., 2024).

Two observations follow. First, once reference/regularization differences are removed, DPO, SimPO, and CPO behave similarly, supporting the hypothesis that much of the reported performance spread across xPOs is driven by a small set of regularizers rather than fundamentally different optimization targets. This is consistent with prior findings that (a) length normalization and the margin term are the dominant

---

[3]Please see Appendix A.1 for further information.

Table 3: Reference model ablation for DPO ($\beta = 0$) and regularization ablation for SimPO ($|y|, \gamma$) and CPO ($\lambda$).

| model | arc_challenge | hellaswag | mmlu | truthfulqa | winogrande | average |
|---|---|---|---|---|---|---|
| Gemma-DPO | 0.3524 | 0.4776 | 0.2614 | 0.2938 | 0.5943 | 0.3959 |
| Gemma-CPO | 0.3498 | 0.4721 | 0.2551 | 0.2925 | 0.5927 | 0.3925 |
| Gemma-SimPO | 0.3609 | 0.4786 | 0.2695 | 0.3060 | 0.5880 | **0.4006** |
| Gemma-PMO | 0.3737 | 0.4568 | 0.2621 | 0.2987 | 0.6014 | 0.3985 |
| Llama-DPO | 0.3208 | 0.4442 | 0.4414 | 0.2546 | 0.5896 | 0.4101 |
| Llama-CPO | 0.3336 | 0.4538 | 0.4346 | 0.2619 | 0.5983 | 0.4164 |
| Llama-SimPO | 0.3387 | 0.4500 | 0.3945 | 0.2583 | 0.5998 | 0.4083 |
| Llama-PMO | 0.3507 | 0.4500 | 0.4526 | 0.2656 | 0.5912 | **0.4220** |
| Qwen-DPO | 0.4471 | 0.5015 | 0.5983 | 0.2387 | 0.6448 | 0.4861 |
| Qwen-CPO | 0.4078 | 0.5115 | 0.5927 | 0.2546 | 0.6417 | 0.4817 |
| Qwen-SimPO | 0.4471 | 0.5014 | 0.5978 | 0.2387 | 0.6440 | 0.4858 |
| Qwen-PMO | 0.4394 | 0.5023 | 0.5984 | 0.2521 | 0.6417 | **0.4868** |

Table 4: Ablation study of $\alpha$ and $\beta$ on benchmark datasets for Gemma, Llama, and Qwen, reporting accuracy on ARC-Challenge, HellaSwag, MMLU, TruthfulQA, WinoGrande, and the overall average.

| alpha | beta | model | arc | hellaswag | mmlu | truthfulqa | winogrande | average |
|---|---|---|---|---|---|---|---|---|
| 0.5 | 0 | Gemma | 0.3447 | 0.4061 | 0.2378 | 0.2827 | 0.5848 | 0.3712 |
| 1 | 0 | Gemma | 0.3558 | 0.4235 | 0.2537 | 0.2925 | 0.5927 | 0.3837 |
| 0.05 | 0.05 | Gemma | 0.3737 | 0.4568 | 0.2621 | 0.2987 | 0.6014 | **0.3985** |
| 0.9 | 0.1 | Gemma | 0.3345 | 0.4119 | 0.2553 | 0.2852 | 0.6077 | 0.3789 |
| 0.95 | 0.05 | Gemma | 0.3430 | 0.4094 | 0.2493 | 0.2840 | 0.5872 | 0.3746 |
| 0.5 | 0 | Llama | 0.3251 | 0.4551 | 0.4496 | 0.2656 | 0.5935 | 0.4178 |
| 1 | 0 | Llama | 0.3294 | 0.4525 | 0.4457 | 0.2668 | 0.6006 | 0.4190 |
| 0.05 | 0.05 | Llama | 0.3507 | 0.4500 | 0.4526 | 0.2656 | 0.5912 | **0.4220** |
| 0.9 | 0.1 | Llama | 0.3251 | 0.4536 | 0.4504 | 0.2619 | 0.5872 | 0.4157 |
| 0.95 | 0.05 | Llama | 0.3396 | 0.4559 | 0.4509 | 0.2668 | 0.5919 | 0.4210 |
| 0.5 | 0 | Qwen | 0.4292 | 0.4996 | 0.5951 | 0.2595 | 0.6346 | 0.4836 |
| 1 | 0 | Qwen | 0.4317 | 0.4991 | 0.5978 | 0.2546 | 0.6361 | 0.4839 |
| 0.05 | 0.05 | Qwen | 0.4394 | 0.5023 | 0.5984 | 0.2521 | 0.6417 | **0.4868** |
| 0.9 | 0.1 | Qwen | 0.4428 | 0.5037 | 0.5972 | 0.2485 | 0.6330 | 0.4850 |
| 0.95 | 0.05 | Qwen | 0.4437 | 0.5021 | 0.5968 | 0.2534 | 0.6267 | 0.4845 |

contributors to SimPO's empirical advantage (Meng et al., 2024), and (b) SimPO (reference-free, length-normalized, marginized) can be understood as a limit or mixture within a length-normalized DPO family, while implementations expose the same knobs (e.g., SimPO-gamma, loss type) under a unified trainer. Second, PMO is competitive or best across backbones under the same ablations, indicating that explicitly matching target probabilities can preserve preference behavior without sacrificing accuracy, even when the distinguishing regularizers in other methods are disabled.

Across both backbones, we observe that reference-based methods (DPO/CDPO/CPO) consistently attain the strongest average performance on the standard academic benchmarks in Figure 5. On Llama3-8B, DPO clearly dominates, achieving the highest average score (0.64) and leading on most individual tasks, while PMO and IPO remain competitive but slightly behind. On Qwen3-8B, PMO attains the best overall average (0.61) and the strongest WinoGrande accuracy, whereas DPO and CPO provide slightly better performance on ARC, HellaSwag, and MMLU. In contrast, the reference-free SimPO objective underperforms substantially on Qwen3-8B in this setting, suggesting that its advantages may be architecture- or data-dependent despite its appealing simplicity.

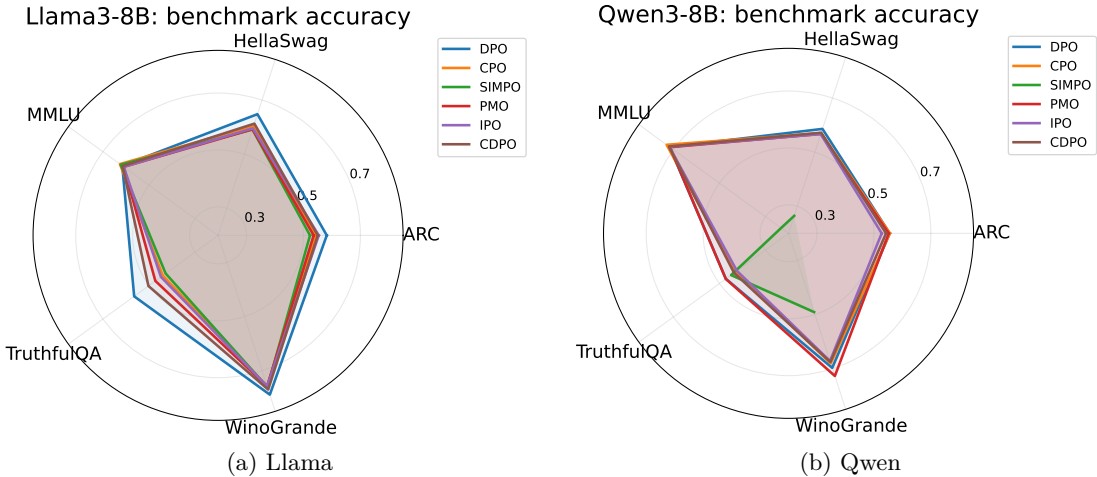

(a) Llama (b) Qwen

Figure 5: Average accuracy (%) on ARC-Challenge, HellaSwag, MMLU, TruthfulQA (MC1), and WinoGrande for CDPO, DPO, IPO, SimPO, CPO, and PMO on (a) Llama, (b) Gemma.

## 6.3 Ablation on the Score Scaling Alpha and Reference Weight Beta

We ablate the PMO hyperparameters that control (i) the strength of the preference signal ($\alpha$ multiplies the pairwise scores) and (ii) the influence of the reference model ($\beta$ scales the reference term, with $\beta=0$ being reference-free). Table 4 summarizes results on ARC-Challenge, HellaSwag, MMLU, TruthfulQA (MC1), and WinoGrande.

**Global trends.** (i) Moving from $\alpha=0.5$ to $\alpha=1.0$ consistently helps or holds steady across backbones at $\beta=0$, indicating that moderately stronger preference signals are beneficial without a reference constraint. (ii) A small but nonzero $\beta$ can further improve accuracy when it is not paired with a too-aggressive $\alpha$. (iii) Over-regularizing the reference (larger $\beta$) together with high $\alpha$ can degrade performance on some backbones, suggesting that the combination of a strong prior and a strong preference signal can oversmooth or miscalibrate the update.

Across backbones, $\alpha$ chiefly governs learning strength and should be set moderately high in the reference-free regime. A small $\beta$ can help, but only when it remains light relative to $\alpha$. Over-regularization (high $\beta$) coupled with aggressive scaling (high $\alpha$) tends to underperform. These findings align with the broader observation that reference-free preference optimization is a strong baseline, and that careful, minimal use of reference regularization can provide incremental, backbone-dependent gains without inducing over-smoothing.

## 7 Conclusion

We introduce a *pure semantic preference scenario* to discuss the preference and accuracy tradeoffs for PMO and other baselines. Across the literature, preference optimization often improves truthfulness and reading comprehension while largely retaining general knowledge, but it can degrade performance on reasoning-heavy math benchmarks unless care is taken in the objective and tuning. This reflects a Pareto-style tension: pushing harder on preference alignment can induce overconfidence or length/format biases that help conversational quality yet erode structured reasoning accuracy. When all benchmarks are reasoning tasks, our PMO, designed for preference alignment, preserves preference adherence without incurring a performance drop on these reasoning evaluations. Our contribution is a controlled analysis of probabilistic preference matching and a simple objective (PMO) with a closed-form solution that allows explicit control of the accuracy–collapse trade-off.

Our work is not without limitations. Due to computational constraint, the experiments are not scaled up to larger models and on-policy algorithms are not further analyzed. Besides, the analysis of the length-variance scenario is also relevant in our pure semantic preference scenario. We leave these to our future work.

## Ethics Statement

This paper presents work whose goal is to advance the field of Machine Learning. There are many potential societal consequences of our work, none of which we feel must be specifically highlighted here.

## Reproducibility Statement

Our code is built on the open-sourced platform OpenRLHF, and we have uploaded the code and synthetic dataset as supplementary files. We will set our repository to public once this paper has been accepted.

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

## The Use of LLMs

The authors used LLMs only for proofreading, checking grammar, and correcting typos to improve the readability of the paper.

## A Additional Experimental Details

### A.1 Comprehensive Descriptions for Models and Datasets

**Models.** In our study, we employ three widely-used open-source large language models to investigate the calibration issue and validate the effectiveness of our proposed method. They include Here's a brief introduction to each model you're using:

- Gemma-3-1B (Google) (Gemma, 2025): A lightweight, open model from the Gemma 3 family; multimodal (accepts text and images) with text output. The 1B size supports a 32K token input context, while larger sizes go to 128K. Gemma 3 emphasizes broad multilingual support (pretrained on 140+ languages) and efficient deployment on limited hardware.

- Llama-3.2-1B (Meta; often shortened to "Llama-3-1B") (Grattafiori & et al., 2024): A 1.23B-parameter, text-only model optimized for multilingual dialogue and on-device use. It supports a 128K token context window, has instruction-tuned variants, and is designed for summarization, rewriting, and agentic tasks. Llama 3.2 targets edge deployment and is optimized for Arm, with day-one enablement on Qualcomm and MediaTek hardware.

- Qwen-2.5-1.5B (Alibaba) (Yang et al., 2024): A 1.54B-parameter model available in base and instruction-tuned variants. The series improves instruction following, coding, math, and structured outputs. The 1.5B models support a 32K token context (with the Instruct variant commonly using up to 8K generation) and multilingual coverage across 29+ languages.

**Benchmark.** To evaluate the efficacy of our proposed calibration method, we employ five datasets to conduct comprehensive experiments:

- ARC-Challenge (Clark et al., 2018): A multiple-choice benchmark of 7,787 grade-school science questions split into Easy and Challenge sets; the Challenge split contains questions that defeat simple retrieval and co-occurrence methods, emphasizing knowledge and reasoning beyond surface cues.

- HellaSwag (Zellers et al., 2019): A commonsense inference dataset where models must choose the most plausible continuation to a context; built via adversarial filtering to be easy for humans (>95% accuracy) yet challenging for models (<48% at release).

- MMLU (Hendrycks et al., 2021): A massive multitask multiple-choice benchmark spanning 57 subjects (humanities, social sciences, STEM, etc.), designed to assess broad world knowledge and problem-solving ability in language models.

- TruthfulQA (MC1) (Lin et al., 2022): Evaluates whether models provide truthful answers to questions targeting common misconceptions; MC1 is the single-correct-option multiple-choice setting (one true answer among 4–5 choices).

- WinoGrande (Sakaguchi et al., 2021): A 44k-instance adversarial Winograd-style pronoun/coreference benchmark with AfLite debiasing to reduce dataset-specific artifacts; improves scale and hardness relative to WSC and supports transfer to related commonsense tasks.

## A.2   The Details of the Synthetic Dataset

**Schema.**   Each example is a quadruple $(x,\ y_A,\ y_B,\ p)$, $p \in (0,1)$, where $x$ is the shared prompt, $y_A, y_B$ are completions differing in exactly one lexical item at the same position, and $p$ is the dataset-specified probability that $y_A$ is preferred. By construction, $(1-p)$ is the probability that $y_B$ is preferred.

In Figure 1, $p = 0.879051$ denotes the target probability of preference given the shared prompt.

**Intended learning target.**   Let $P_\theta(y_A \succ y_B|x)$ denote the model's pairwise preference probability under a Bradley–Terry–style parameterization (Li et al., 2023):

$$P_\theta(y_A \succ y_B|x) \;=\; \frac{\exp(r_\theta(x, y_A))}{\exp(r_\theta(x, y_A)) + \exp(r_\theta(x, y_B))},$$

where $r_\theta$ is a scalar scoring function. Our dataset defines a target soft label $p$ for this pairwise probability. Thus, the alignment goal is probability matching, i.e., $P_\theta(y_A \succ y_B|x) \approx p$ over the distribution of minimal pairs. This soft-preference formulation generalizes binary chosen–rejected labels used in standard preference datasets (Rafailov et al., 2023; Li et al., 2023) by supplying calibrated targets for pairwise comparisons.

**Generation and controls.**   To construct $(x, y_A, y_B)$ we: (a) sample a prompt template that admits a single-slot substitution; (b) choose a lexical contrast set $\{w_A, w_B\}$ (e.g., brand, beverage, team, OS) and instantiate $y_A, y_B$ by substituting $w_A$ vs. $w_B$ in the same position; (c) verify minimality (string equality outside the substituted span) and well-formedness. This process controls for length, formatting, and syntactic variation, leaving only the targeted semantic contrast to influence model preferences (Warstadt et al., 2020). The probability $p$ is then sampled by a fixed-seed RNG and stored with the pair.

# B   Additional Comparison with Preference Alignments Objectives

## B.1   Optimal Policy of Variants of DPO

**Optimal policies of DPO and PPO.**   The optimal policy of DPO is given by

$$\pi^\star(y|x) = \frac{\pi_{\text{ref}}(y|x) \exp(r(x, y)/\beta)}{\sum_{y'} \pi_{\text{ref}}(y'|x) \exp(r(x, y')/\beta)}, \tag{9}$$

as it is discussed in the main text. PPO is widely used for RLHF. Since our goal is not to analyze PPO's convergence properties, we instead adopt the RLHF optimal policy (Equation 9) as a proxy for the PPO solution.

**Optimal solution of cDPO.**   The optimal policy of cDPO is given by

$$\pi^\star(y|x) = \frac{\pi_{\text{ref}}(y|x) \exp(c\,\pi_\phi(y|x))}{\sum_y \pi_{\text{ref}}(y|x) \exp(c\,\pi_\phi(y|x))}, \qquad c = \frac{1}{\beta}\log\frac{1-\varepsilon}{\varepsilon}$$

**Optimal policy of IPO.**   The optimal policy of cDPO is given by

$$\pi^\star(y|x) = \frac{\pi_{\text{ref}}(y|x)\exp\left(\frac{1}{\beta}\,\mathbb{E}_{y'\sim\mu}[p^*(y \succ y'|x)]\right)}{\sum_y \pi_{\text{ref}}(y|x)\exp\left(\frac{1}{\beta}\,\mathbb{E}_{y'\sim\mu}[p^*(y \succ y'|x)]\right)}.$$

**Optimal policy of NashMD.**   NashMD is used to optimize the objective of Nash learning from human feedback (NLHF). Since our goal is not to analyze convergence properties, we adopt the (unknown) NLHF optimal policy as a surrogate for the NashMD solution.

To the best of our knowledge, NLHF admits no closed-form optimal policy. The strongest available characterization shows that the NLHF Nash equilibrium coincides with the solution of online IPO, which can be expressed in the following recursive form:

$$\pi^\star(y|x) = \frac{\pi_{\text{ref}}(y|x)\exp\left(\frac{1}{\beta}\,\mathbb{E}_{y'\sim\pi^\star(y|x)}[p^*(y \succ y'|x)]\right)}{\sum_y \pi_{\text{ref}}(y|x)\exp\left(\frac{1}{\beta}\,\mathbb{E}_{y'\sim\pi^\star(y|x)}[p^*(y \succ y'|x)]\right)}.$$

**Optimal policy of CPO.** The objective of CPO is given by

$$-\log \sigma(\beta \log \pi_\theta(y_w \mid x) - \beta \log \pi_\theta(y_l \mid x)) - \lambda \log \pi_\theta(y_w \mid x),$$

which is originated from a constraint optimization problem

$$\min - \log \sigma(\beta \log \pi_\theta(y_w \mid x) - \beta \log \pi_\theta(y_l \mid x)) \quad s.t. \quad \log \pi_\theta(y_w \mid x) \leq \epsilon.$$

With out the constraint (or $\lambda = 0$), the loss function admits the following closed form solution.

$$\pi^\star(y|x) = \frac{\exp\left(\frac{1}{\beta}r(x,y)\right)}{\sum_y \exp\left(\frac{1}{\beta}r(x,y)\right)}$$

With the constraint, there is generally no closed form solution.

## B.2 Comparison with H-DPO

Omura et al. (2024) introduced a variant of DPO, termed H-DPO. By decomposing the reverse KL divergence into its entropy and cross-entropy components, one can separately adjust the entropy contribution through a parameter $\alpha$. The resulting objective for entropy-adjusted DPO is

$$
\begin{aligned}
J_{\text{H-DPO}} &= \mathbb{E}_{x\sim\mathcal{D},\, y\sim\pi}\left[r(x,y) - \beta\, D_\alpha\big(\pi \,\|\, \pi_{\text{ref}}\big)\right] \\
&= \mathbb{E}_{x\sim\mathcal{D},\, y\sim\pi}[r(x,y)] + \alpha\beta\, H(\pi) - \beta\, H\big(\pi, \pi_{\text{ref}}\big).
\end{aligned}
$$

While both PMO and H-DPO incorporate an entropy term, their underlying principles differ. In PMO, the final term is a KL divergence, whereas in H-DPO the final term is a cross-entropy; in fact, the combination of the second and third terms in H-DPO recovers the KL divergence.

# C  Technical Results

## C.1  Property C.1

Let the PMF of $p_i$ is $f(x), x \in [0,1]$, $i = 1, \cdots, n$.

$$\text{PCI} = 2 \int_0^{0.5} f(x)dx. \tag{10}$$

**Property C.1 (PCI consistency)** *(a) Consistency: by the law of large numbers,* $\text{PCI}_n \to \mathbb{E}[\min(P, 1-P)]$ *almost surely as* $n \to \infty$. *(b) Tight bounds:* $0 \leq \mathbb{E}[\min(P, 1-P)] \leq \frac{1}{2}$. *The lower bound is attained when* $P \in \{0,1\}$ *a.s.; the upper bound is attained when* $P \equiv \frac{1}{2}$ *a.s.*

Proof sketch. (a) Apply the strong law to the i.i.d. sequence $\min\{p_i, 1-p_i\}$. (b) Pointwise, $0 \leq \min(p, 1-p) \leq 1/2$; take expectations and note the extremal cases. (c) Use LOTUS to write $\mathbb{E}[\min(P, 1-P)] = \int \min(x, 1-x)f(x)\,dx$ and split at $1/2$; alternatively use the tail integral $\int_0^{1/2} \mathbb{P}(\min(P, 1-P) > t)\,dt = \int_0^{1/2}(F(1-t) - F(t))\,dt$. (d) Follows immediately from (c) under symmetry. (e) For $U \sim \text{Unif}(0,1)$, $\mathbb{P}(\min(U, 1-U) \leq t) = 1 - \mathbb{P}(U \in [t, 1-t]) = 2t$ on $t \in [0, 1/2]$, giving the stated distribution and mean $1/4$.

## C.2  Proof of Proposition 4.1

Consider the RL problem:

$$\max_\phi \ \mathbb{E}_{x\sim\rho}\, \mathbb{E}_{y\sim\pi_\phi(\cdot|x)}\Big[r_\gamma(x,y)\Big] + \beta' H\big(\pi_\phi(\cdot|x)\big).$$

It can be written as

$$\min_\phi \mathbb{E}_{x\sim\rho}\mathbb{E}_{y\sim\pi_\phi(\cdot|x)} \log \pi_\phi(y|x) - \log\left[\exp(\frac{1}{\beta'}r_\gamma(x,y))\right].$$

The optimal solution is

$$\pi_\phi(y|x) = \frac{1}{Z(x)} \exp(\frac{1}{\beta'} r_\gamma(x,y)),$$

where

$$Z(x) = \sum_y \exp(\frac{1}{\beta'} r_\gamma(x,y)).$$

This gives the second result in Proposition 4.1: the optimal policy is given by

$$\pi^\star(y|x) = \exp\left(\frac{1}{\beta'} r_\gamma(x,y)\right) / \sum_{y'} \exp\left(\frac{1}{\beta'} r_\gamma(x,y')\right).$$

The reward can be written as

$$r_\gamma(x,y) = \beta' \log \pi_\phi(y_w|x) + \beta' \log Z(x).$$

Put it into the loss function of reward with margin $\gamma$, which is

$$-\mathbb{E}_{(x,y_w,y_l)} \log \sigma\left(r(x,y_w) - r(x,y_l) - \gamma\right).$$

we obtain the objective SimPO:

$$- \mathbb{E}_{(x,y_w,y_l)} \log \sigma\left(\beta' \log \pi_\phi(y_w|x) - \beta' \log \pi_\phi(y_l|x) - \gamma\right)$$
$$= - \mathbb{E}_{(x,y_w,y_l)} \log \sigma\left(\frac{\beta}{|y_w|} \log \pi_\phi(y_w|x) - \frac{\beta}{|y_l|} \log \pi_\phi(y_l|x) - \gamma\right).$$

### C.3 Proof of Proposition 4.2

Consider the problem:

$$\max_\phi \mathbb{E}_{x\sim\rho}\mathbb{E}_{y\sim\pi_\phi(\cdot|x)} r(x,y) + \alpha H(\pi_\phi(y|x)) - \beta D_{\mathrm{KL}}(\pi_\phi(y|x)\|\pi_{\mathrm{ref}}(y|x)), \tag{11}$$

Equation (11) can be written as

$$\min_\phi \mathbb{E}_{x\sim\rho}\mathbb{E}_{y\sim\pi_\phi(\cdot|x)} \log \pi_\phi(y|x) - \log\left[\pi_{\mathrm{ref}}(y_l|x)^{\frac{\beta}{\alpha+\beta}} \exp(\frac{1}{\alpha+\beta} r(x,y))\right],$$

The optimal solution is

$$\pi_\phi(y|x) = \frac{1}{Z(x)} \pi_{\mathrm{ref}}(y_l|x)^{\frac{\beta}{\alpha+\beta}} \exp(\frac{1}{\alpha+\beta} r(x,y)),$$

where

$$Z(x) = \sum_y \pi_{\mathrm{ref}}(y_l|x)^{\frac{\beta}{\alpha+\beta}} \exp(\frac{1}{\alpha+\beta} r(x,y)).$$

The reward can be written as

$$r(x,y) = (\alpha+\beta) \log \frac{\pi_\phi(y_w|x)}{\pi_{\mathrm{ref}}(y_w|x)^{\frac{\beta}{\alpha+\beta}}} + (\alpha+\beta) \log Z(x).$$

Put it into the loss function of reward, we obtain the DPO version:

$$-\mathbb{E}_{(x,y_w,y_l)} \log \sigma\left((\alpha+\beta) \log \frac{\pi_\phi(y_w|x)}{\pi_{\mathrm{ref}}(y_w|x)^{\frac{\beta}{\alpha+\beta}}} - (\alpha+\beta) \log \frac{\pi_\phi(y_l|x)}{\pi_{\mathrm{ref}}(y_l|x)^{\frac{\beta}{\alpha+\beta}}}\right).$$

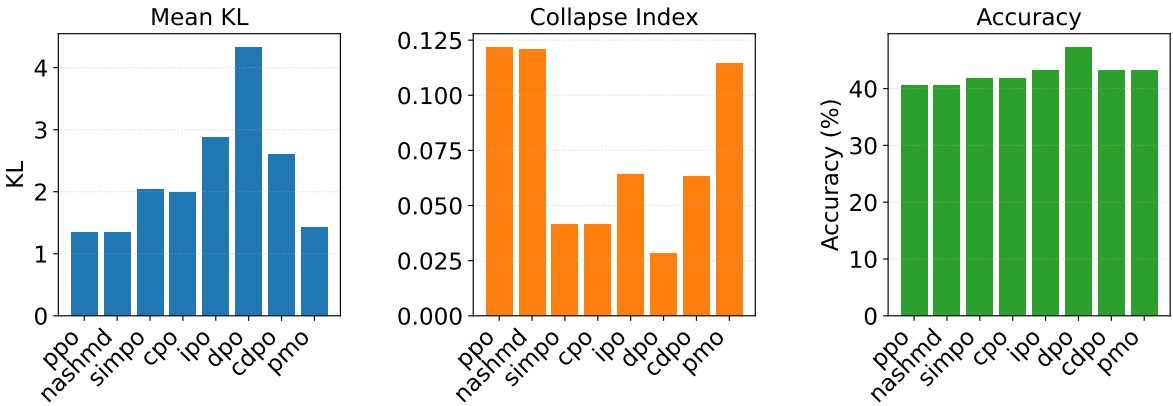

Figure 6: Tradeoff between preference and accuracy on Gemma model.

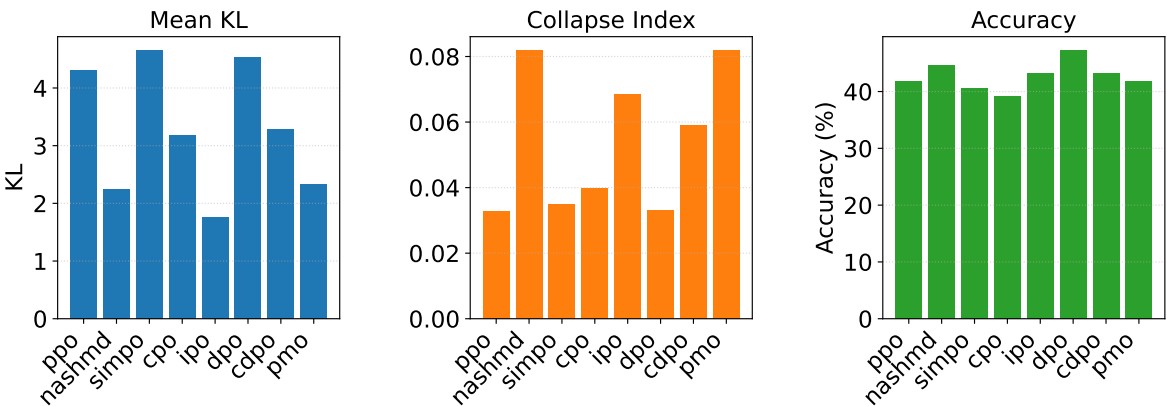

Figure 7: Tradeoff between preference and accuracy on Qwen model.

# D  Additional Experiment

## D.1  Backbone-Wise Observations in Synthetic Dataset

**Qwen.**  IPO attains the lowest KL (1.76), followed by Nash-MD (2.24) and PMO (2.32), indicating better alignment of predicted probabilities with ground-truth targets. However, the highest accuracy is delivered by DPO (0.473), which exhibits the second-worst KL (4.53) and one of the lowest PCI values (0.033), i.e., strong collapse. PPO and SimPO also show relatively low PCI ($\approx$0.033–0.035) with large KL ($\approx$4.30–4.66). In contrast, methods with higher PCI (less collapse), such as PMO and Nash-MD (PCI $\approx$0.082), tend to have lower KL but slightly lower accuracy (0.419 and 0.446, respectively). Overall, for Qwen we observe a clear trade-off: pushing accuracy via more decisive predictions (lower PCI) correlates with worse KL, suggesting overconfidence that increases divergence when predictions are wrong.

**Gemma.**  The best KL is achieved by Nash-MD (1.35) and PPO (1.35), both with the highest PCI ($\approx$0.121), i.e., least collapse. DPO again yields the highest accuracy (0.473) but the worst KL (4.33) and the lowest PCI (0.029), signaling pronounced collapse. CDPO, IPO, and PMO occupy intermediate positions: their KL is higher than PPO/Nash-MD but lower than DPO; their PCI is below PPO/Nash-MD but above DPO. This backbone thus strengthens the pattern that methods achieving better probabilistic alignment (low KL) do so by avoiding extreme confidence (higher PCI), whereas the most accurate method (DPO) concentrates probability mass aggressively (very low PCI), incurring high KL.

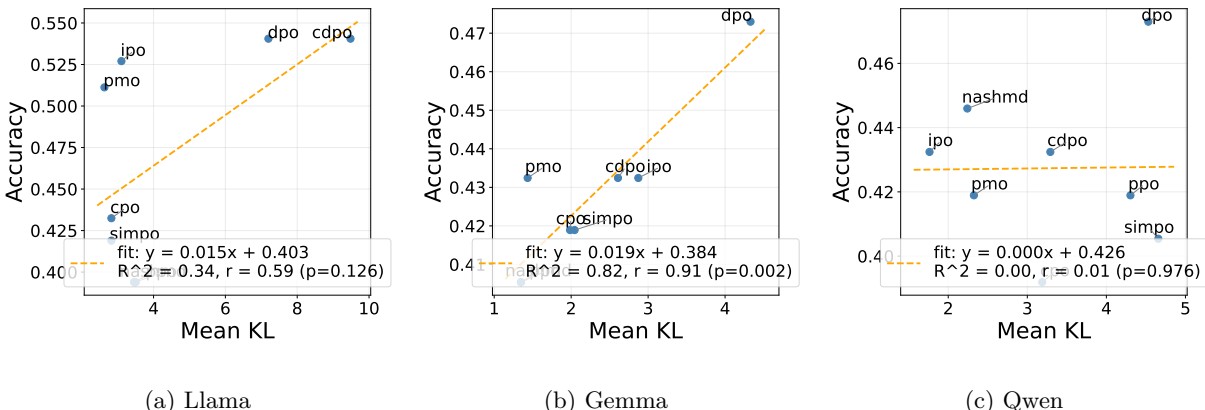

(a) Llama      (b) Gemma      (c) Qwen

Figure 8: Linear regressions of KL-Accuracy for Llama, Gemma, and Qwen.

Table 5: Summary of Llama3-8B results across optimization algorithms.

| Algorithm | KL ↓ | PCI ↑ | Accuracy ↑ |
|-----------|--------|--------|------------|
| DPO | 2.1173 | 0.1284 | 0.4583 |
| CPO | 1.8208 | 0.1160 | **0.5278** |
| SimPO | **1.7322** | 0.1207 | 0.4722 |
| PMO | 1.8749 | **0.1327** | 0.4722 |
| IPO | 2.1659 | 0.1239 | 0.4444 |
| CDPO | 2.1511 | 0.1282 | 0.4306 |

### D.2 Analysis of KL-PCI-Accuracy Index

Across backbones, linear regressions of PCI on KL reveal a strong, negative association: Qwen ($r = -0.888$, $p$=0.0032, slope $-0.017 \pm 0.0036$), Gemma ($r = -0.754$, $p$=0.0308, slope $-0.0286 \pm 0.0102$), and Llama ($r = -0.860$, $p$=0.0061, slope $-0.00384 \pm 0.00093$). Thus, as KL increases, PCI systematically decreases, i.e., higher divergence correlates with stronger collapse. Spearman's $\rho$ is also negative and significant for Qwen and Gemma, while Llama shows a strong linear trend but weaker rank monotonicity.

Across the three backbones, the relationship between KL (treated as a distance to the target distribution) and accuracy differs markedly. For Qwen, there is essentially no association: Pearson $r = 0.013$ ($p = 0.976$), a near-zero slope ($0.00029 + / - 0.00908$), and $R^2 \approx 0$, indicating accuracy is insensitive to KL within this range. Gemma shows a strong, statistically significant positive correlation ($r = 0.906$, $p = 0.002$; Spearman $\rho = 0.84$, $p = 0.009$) with a tight linear fit ($R^2 = 0.82$, RMSE $= 0.0086$): the slope ($0.0192 + / - 0.0037$; 95% CI [0.010, 0.028]) implies each unit increase in KL aligns with 1.9 percentage-point higher accuracy, evidencing a pronounced tradeoff where higher divergence from the target probabilities accompanies better task accuracy. Llama exhibits a similar positive trend ($r = 0.587$; slope $0.0153 + / - 0.0086$) but it is not statistically significant at $n = 8$ ($p = 0.126$; CI spans zero), and rank association is weak ($\rho = 0.313$, $p = 0.450$). In short: no KL–accuracy tradeoff for Qwen, a clear positive tradeoff for Gemma, and an inconclusive trend for Llama, with small-sample uncertainty cautioning interpretation.

## E Additional Results on Benchmark

For Llama3-8B, we observe a clear trade-off between alignment quality and output diversity. Among all methods, SimPO achieves the lowest KL divergence (1.73), while PMO attains the highest PCI (0.133), indicating stronger resistance to collapse at comparable KL levels. In contrast, CPO yields the best accuracy (0.528) but with slightly higher KL than SimPO, suggesting that peak task performance does not always coincide with the most stable KL–PCI balance. Overall, reference-free objectives (especially SimPO) remain

Table 6: Algorithm-wise average accuracy on ARC-Challenge, HellaSwag, MMLU, TruthfulQA, and Wino-Grande for DPO, CPO, SimPO, PMO, IPO, and cDPO on Llama-3-8B and Qwen-3-8b.

| Model | arc_challenge | hellaswag | mmlu | truthfulqa | winogrande | average |
|---|---|---|---|---|---|---|
| Llama3-DPO | **0.5819** | **0.6478** | 0.6161 | **0.5644** | **0.7885** | **0.6397** |
| Llama3-CPO | 0.5435 | 0.6045 | **0.6263** | 0.4403 | 0.7569 | 0.5943 |
| Llama3-SimPO | 0.5230 | 0.5912 | 0.6226 | 0.4278 | 0.7609 | 0.5851 |
| Llama3-PMO | 0.5358 | 0.5920 | 0.6096 | 0.4724 | 0.7577 | 0.5935 |
| Llama3-IPO | 0.5538 | 0.5955 | 0.6097 | 0.4481 | 0.7569 | 0.5928 |
| Llama3-cDPO | 0.5503 | 0.6128 | 0.6144 | 0.5025 | 0.7695 | 0.6099 |
| Qwen3-DPO | 0.5555 | **0.5858** | 0.7160 | 0.4720 | 0.6969 | 0.6053 |
| Qwen3-CPO | **0.5563** | 0.5705 | **0.7294** | 0.4309 | 0.6803 | 0.5935 |
| Qwen3-SimPO | 0.1869 | 0.2646 | 0.2321 | 0.4495 | 0.4909 | 0.3248 |
| Qwen3-PMO | 0.5520 | 0.5673 | 0.7149 | **0.4725** | **0.7269** | **0.6067** |
| Qwen3-IPO | 0.5273 | 0.5666 | 0.7176 | 0.4242 | 0.6717 | 0.5815 |
| Qwen3-cDPO | 0.5427 | 0.5717 | 0.7195 | 0.4364 | 0.6756 | 0.5892 |

competitive on accuracy while offering a favorable KL–PCI trade-off, consistent with prior findings that simpler reference-free formulations can reduce optimization sensitivity and complexity (Meng et al., 2024).

