# OpenReview forum: "On Preference Optimization in Large Language Models Under Pure Semantic Preferences"
_TMLR — Rejected by TMLR_

### Review · Reviewer_HUxj · 2026-01-20

**Summary Of Contributions:**

This work constructs a pure semantic preference setting for a idealized evaluation of RLHF and DPO-family models by ruling out the length and syntactic bias.  The evaluation reveals that most alignment methods still fail to fully capture the human "semantic" preference. Then, a lightweight preference-matching optimization is proposed to suit the pure semantic setting, which is demonstrated effective in both idealized and practical settings.

The paper makes two contributions:

i) It decomposes human preference gaps into length, syntactical and semantic gaps, and accordingly presents a pure semantic preference scenario for studying the capability of existing alignment methods in capturing the human semantic preferences;

ii) It proposes a lightweight PMO algorithm for addressing the limitations of existing alignment methods on the pure semantic setting.

Its key strengths are:
i) A novel perspective that decomposes human preference gaps into sub-gaps, isolating syntactical, length bias from semantic gap.

ii) A systematic evaluation of human alignment methods on the pure semantic preference setting, revealing their insufficiency in learning from human semantic preferences.

iii) A novel algorithm designed for the pure semantic setting that also works well in practical scenarios.

Its key weaknesses are:
i) The decomposition of human preference gaps implicitly assume that length, syntactics and semantics correspond to mutually independent dimensions of gaps. However, they are usually correlated with each other.

ii) The decomposition does not effectively lead to an analysis of each of the three types of gaps, but merely motivates construction of the benchmark with paired samples of same lengths and syntactical structures.

iii) The proposed PMO objective lacks novelty in that it can be seen as a reweighting of $\pi_\phi$ and $\pi_{ref}$.

**Audience:**

Yes

**Audience Explanation:**

It targets at human alignment, a very popular topic in the LLM community.

**Broader Impact Concerns:**

Not as far as I am concerned.

**Claims And Evidence:**

Yes

**Claims Explanation:**

The experiment demonstrates the effectiveness of the proposed off-policy PMO variant.

**Requested Changes:**

Please refer to the weaknesses above. Clarifying my concerns are crucial to my recommendation for acceptance.

---

> ### Author Response · Authors · 2026-03-17
> **Rebuttal**
>
> **1) The decomposition of human preference gaps implicitly assume that length, syntactics and semantics correspond to mutually independent dimensions of gaps. However, they are usually correlated with each other.**
>
>
> Thank you very much for pointing out an important perspective that length, syntactics and semantics are correlated. To address this issue, we have redefined the definition of these gaps as the residual gap of the previous alignment. We state it as a working hypothesis in our paper as follow.
>
> [Orthogonal Attribution of Alignment]
> We assume that the total alignment gap $G(y_1, y_2) = \log P_\theta(y_1 \mid x) - \log P_\theta(y_2 \mid x)$ can be decomposed into a sequence of additive marginal contributions by traversing a path of intermediate latent representations:
>
> 1. There exists an intermediate response $y_2''$ that shares the same meaning as $y_2$ but matches the length of $y_1$. The \textbf{length gap} is defined as $\log P_\theta(y_2'' \mid x) - \log P_\theta(y_2 \mid x)$.
>
> 2. There exists an intermediate response $y_2'$ that shares the same meaning as $y_2''$ but matches both the length and syntactic structure of $y_1$. The \textbf{syntax gap}, although naturally correlated with length, is defined as the residual gap $\log P_\theta(y_2' \mid x) - \log P_\theta(y_2'' \mid x)$ conditioned on the fixed length.
>
> 3. The **semantic gap** is defined as the residual gap $\log P_\theta(y_1 \mid x) - \log P_\theta(y_2' \mid x)$ conditioned on both the length and syntactic alignment.
>
>
> **2) The decomposition does not effectively lead to an analysis of each of the three types of gaps, but merely motivates construction of the benchmark with paired samples of same lengths and syntactical structures.**
>
> Thank you for the question. While we agree that a primary utility of the proposed decomposition is to motivate the construction of benchmarks with controlled length and syntactic structure, its significance extends to the diagnostic analysis of preference alignment. By isolating specific components of the alignment gap, the framework enables a fine-grained attribution of model bias that aggregate preference scores fail to capture. For instance, it allows for the quantification of ``length bias'' ($\Delta_{\mathrm{len}}$) independently of semantic merit, providing a rigorous metric to evaluate whether specific alignment algorithms, such as DPO or RLHF, disproportionately reward verbosity or syntactic canonicality. Furthermore, by isolating the residual semantic gap ($\Delta_{\mathrm{sem}}$) on controlled pairs, one can identify systematic failures in a model's preference logic that would otherwise be obscured by surface-level features. Thus, the decomposition serves as an analytical lens to inspect the underlying drivers of model preferences, offering diagnostic insights beyond the scope of standard balanced benchmarks. We leave more extensive empirical applications of this analysis to future work.
>
>
> **3) The decomposition does not effectively lead to an analysis of each of the three types of gaps, but merely motivates construction of the benchmark with paired samples of same lengths and syntactical structures.**
>
> Thank you for pointing out.
> Note that by substituting the KL divergence with the cross-entropy term in Equation 7, the objective recovers the SPL objective \citep{SlocumPH25}. Specifically, the two objectives become identical under the reparameterization $\alpha \rightarrow \alpha - \beta$. In this sense, PMO is not presented as a fundamentally new objective function, but rather as a specialized formulation designed for the controlled semantic preference setting and the subsequent diagnostic analysis of probability matching versus collapse. Nevertheless, our framework emphasizes a parameter regime where the coefficient of the entropy term remains strictly positive relative to the KL term—a distinction not prioritized in the SPL literature. We retain the name PMO to highlight this specific focus and to maintain consistency with the reference-attenuation perspective established in Proposition 4.2.

---

### Review · Reviewer_V628 · 2026-02-10

**Summary Of Contributions:**

The paper introduces a "pure semantic preference" setting where response pairs differ by a single word with soft probabilistic labels, isolating semantic preference from length/syntax confounds. The authors evaluate existing alignment methods in this setting, observe an accuracy-calibration tradeoff, and propose Preference Matching Optimization (PMO), which combines entropy and KL regularization with a closed-form optimum. Experiments use three 1B-scale models on synthetic and standard benchmarks.

**Additional Comments:**

I'd encourage the authors to lean into the diagnostic/analytical angle (showing what happens to existing methods under controlled semantic conditions) rather than framing PMO as the primary contribution.

**Audience:**

Yes

**Audience Explanation:**

The question of how alignment methods behave when non-semantic confounds are removed is genuinely interesting, and the empirical observations (on-policy > off-policy, reference-free methods better preserve calibration) are practically relevant. The pure semantic evaluation framework, while limited in its current form, could inspire more rigorous controlled evaluations of preference optimization methods.

**Claims And Evidence:**

No

**Claims Explanation:**

The central methodological claim (that PMO is a novel objective) is undermined by its equivalence to SPL (Slocum et al., ICLR 2025), which is not cited. The synthetic experiments use randomly generated preference labels rather than meaningful semantic preferences, weakening the "pure semantic" narrative.

**Requested Changes:**

1- Cite and position relative to Soft Preference Learning (SPL). The PMO objective is essentially identical to SPL from Slocum et al. (2025). SPL proposes the same entropy/cross-entropy decoupling, derives the same closed-form optimal policy, and shows RLHF/DPO are special cases. The motivating contexts differ (semantic isolation vs. diversity/social choice), but the method is the same. PMO must be repositioned as an application of this objective to the controlled semantic setting, not as a novel algorithm.

2- All figures and tables show single-run numbers. Given the small margins between methods, statistical significance cannot be assessed

3- Justify or replace the random soft labels. The synthetic preference labels are drawn from a random number generator, meaning models are memorizing arbitrary numbers rather than learning meaningful semantic preferences. Consider deriving labels from human judgments, LLM judges, or at minimum, distributions grounded in real preference patterns

4- Scale experiments beyond 1B models. All results use 1B–1.5B parameter models. At least one 7B experiment would substantially increase confidence in the findings

---

> ### Author Response · Authors · 2026-03-17
> **Rebuttal**
>
> **1) Positioning vs. Soft Preference Learning (SPL): “PMO objective is essentially identical”**
>
> Thank you for pointing out.
> Note that by substituting the KL divergence with the cross-entropy term in Equation 7, the objective recovers the SPL objective \citep{SlocumPH25}. Specifically, the two objectives become identical under the reparameterization $\alpha \rightarrow \alpha - \beta$. In this sense, PMO is not presented as a fundamentally new objective function, but rather as a specialized formulation designed for the controlled semantic preference setting and the subsequent diagnostic analysis of probability matching versus collapse. Nevertheless, our framework emphasizes a parameter regime where the coefficient of the entropy term remains strictly positive relative to the KL term—a distinction not prioritized in the SPL literature. We retain the name PMO to highlight this specific focus and to maintain consistency with the reference-attenuation perspective established in Proposition 4.2.
>
> Manuscript added in Sec. 1
>
> `This objective is equivalent to Soft Preference Learning (SPL; \citep{SlocumPH25}) up to a simple reparameterization; we build on this objective but study it through our controlled semantic probability-matching lens and the resulting reference-attenuation interpretation.`
>
> **2) “Single-run numbers only; significance unclear”**
>
> Thank you for the question. While we agree that multi-seed training would be ideal, repeated LLM preference-optimization finetunes are computationally expensive (especially when sweeping methods/hyperparameters), and it is common in contemporary LLM work to report single-seed training runs for each condition. For transparency, we will (i) explicitly mark all reported results as single-run, (ii) avoid claims of statistical significance based on small margins, and (iii) add uncertainty estimates computed without additional training—e.g., confidence intervals / bootstrap intervals over evaluation instances—for Accuracy, Mean-KL, and PCI. This provides a more statistically grounded view of whether observed gaps are robust on the evaluation set, even when multi-seed retraining is not feasible in the current compute budget.
>
> **3) “Random soft labels are unjustified / meaningless; models memorize arbitrary numbers”**
>
> Thank you for the insightful question. We agree that the soft targets 𝑝p are synthetic. This is by design: our goal in the pure semantic preference scenario is to study probability matching after removing non-semantic confounds (length/template/format), where the two candidates are semantically neutral (no notion of “true vs false”) and differ only in a controlled lexical substitution.
>
> In this regime, there is typically no naturally available, scalable “ground-truth” distribution over subjective alternatives (e.g., “tea vs coffee”, “cola vs pepsi”) for each minimal pair; collecting calibrated human preference fractions for every pair would require dedicated annotation and is outside the scope of constructing a strictly controlled diagnostic dataset.
>
> Consequently, we treat p as a dataset-specified target preference probability and sample it with a fixed-seed RNG for reproducibility, explicitly to decouple supervision from incidental stylistic/length artifacts and lexical priors, so that any systematic distortion in predicted probabilities can be attributed to the learning objective rather than to spurious correlations in the data construction.
>
> Finally, because the paired responses are subjective, semantically neutral alternatives (not factual claims or safety-critical content), optimizing to match these distributional targets is a low-stakes probe of whether preference optimization preserves calibrated uncertainty vs. collapses to extremes; it is not intended to teach models arbitrary “semantic facts.”
>
> Manuscript added in Sec. 3
>
> `The value $p$ is synthetically specified (fixed RNG seed for reproducibility) because the pure semantic setting lacks scalable, naturally available preference-frequency targets for subjective minimal pairs; the resulting supervision is a low-stakes probe of probability matching/collapse rather than a claim about semantic “truth”. `

---

> ### Author Response · Authors · 2026-03-17
> **Rebuttal 2**
>
> **4) Scale experiments beyond 1B models. All results use 1B–1.5B parameter models. At least one 7B experiment would substantially increase confidence in the findings**
>
> Thank you for pointing out. We thank the reviewer for this suggestion. We agree that evaluating beyond the 1B–1.5B scale is important for establishing whether the observed trends persist in more realistic deployment regimes. In fact, our revised experiments already include 8B-scale models, specifically Llama3-8B and Qwen3-8B, and we will make this much more explicit in the main text to avoid the impression that the study is limited to 1B-scale backbones.
>
> Specifically, Table 1 reports KL divergence, PCI, and downstream accuracy for multiple preference optimization methods on both Llama3-8B and Qwen3-8B. These results show that the key trends observed at smaller scales continue to hold at 8B scale. On Llama3-8B, SimPO achieves the lowest KL (1.73) while maintaining competitive accuracy, CPO reaches the highest accuracy (0.53) at the cost of somewhat larger KL and reduced PCI, and PMO attains the strongest PCI (0.13). On Qwen3-8B, PMO provides the best KL–PCI trade-off, with the lowest KL (1.66) and highest PCI (0.16), while DPO achieves the best accuracy (0.50) with moderately higher KL. Overall, these 8B results support the same conclusion as in the smaller-scale experiments: reference-free or partially reference-relaxed methods can remain competitive in accuracy while often exhibiting more favorable KL–PCI behavior, suggesting improved robustness against representational collapse.
>
> We also include standard academic benchmark results at 8B scale in Figure 5. Across both backbones, the results again show meaningful separation between methods at larger scale rather than a collapse of the trends seen at 1B. On Llama3-8B, DPO achieves the strongest average score (0.64) and leads on most individual tasks, while PMO and IPO remain competitive. On Qwen3-8B, PMO attains the best overall average (0.61) and the strongest WinoGrande accuracy, whereas DPO and CPO perform slightly better on ARC, HellaSwag, and MMLU. These experiments therefore already provide the “beyond-1B” validation requested by the reviewer.
>
> Regarding model coverage, we did not include Gemma at this scale because there is no directly matched 8B Gemma variant in our setup, so for larger-scale experiments we focused on the two available and widely used 8B backbones, Llama3-8B and Qwen3-8B.
>
> Manuscript added in Sec. 5 and 6
>
> `Trade-offs on 8B models. Table 1 compares KL divergence, PCI, and downstream accuracy for different preference optimization methods on Llama3-8B and Qwen3-8B. On Llama3-8B, SimPO attains the lowest KL (1.73) while maintaining competitive accuracy, whereas CPO reaches the highest accuracy (0.53) at the cost of slightly larger KL and reduced PCI, and PMO yields the strongest PCI (0.13). On Qwen3-8B, PMO offers the best KL–PCI trade-off, achieving the lowest KL (1.66) and the highest PCI (0.16), while DPO delivers the best accuracy (0.50) with moderately higher KL. Overall, reference-free or partially reference-relaxed methods such as SimPO and PMO can match or closely approach the accuracy of DPO/CPO while often providing more favorable KL–PCI profiles, indicating improved robustness against representational collapse.`
>
> `Across both backbones, we observe that reference-based methods (DPO/CDPO/CPO) consistently attain the strongest average performance on the standard academic benchmarks in Figure 5. On Llama3-8B, DPO clearly dominates, achieving the highest average score (0.64) and leading on most individual tasks, while PMO and IPO remain competitive but slightly behind. On Qwen3-8B, PMO attains the best overall average (0.61) and the strongest WinoGrande accuracy, whereas DPO and CPO provide slightly better performance on ARC, HellaSwag, and MMLU. In contrast, the reference-free SimPO objective underperforms substantially on Qwen3-8B in this setting, suggesting that its advantages may be architecture- or data-dependent despite its appealing simplicity.`

---

### Review · Reviewer_qkwW · 2026-02-17

**Summary Of Contributions:**

This work investigates the alignment of Large Language Models with human preferences, specifically focusing on a pure semantic preference scenario. By isolating this setting, the study removes confounding factors such as length bias and syntactic alignment. The authors demonstrate that even in these simplified cases, current methods fail to fully capture underlying preferences. Consequently, they propose Preference-Matching Optimization (PMO), a novel, lightweight method with a closed-form optimum tailored for the semantic domain. Experimental results confirm the efficiency and effectiveness of the PMO algorithm.

**Audience:**

Yes

**Audience Explanation:**

Because the paper addresses a major challenge in LLMs, provides a new algorithm, and isolates specific variables (semantics vs. syntax/length) to provide new insights, it meets the threshold of being interesting to at least a subset of the TMLR audience.

**Claims And Evidence:**

Yes

**Claims Explanation:**

Experimental results confirm the efficiency and effectiveness of the PMO algorithm.

**Requested Changes:**

1. **Clarification of Section 3: Pure Semantic Preference (Critical)**
The definition related to "Pure Semantic Preference" in Section 3 is not clear enough and needs to be clarified:
   * **(a)** What is the definition of $T_i''$ in equation (1)?
   * **(b)** What is the meaning of "further enforce semantic preservation"? According to the definition, $y_2''$ is already a subsequence of $y_2$. Does the author mean we further enforce semantic preservation for it to be $\hat{y}_2$? If this is the case, should it be $m(\hat{y}_2) = m(y_1)$?
   * **(c)** Can you provide more detail about the syntactic similarity? There is also a lack of a definition for the "semantic representation map," though this is a minor issue since vocabulary semantics are relatively fixed. However, syntactic similarity seems much harder to evaluate; please provide more detail.
   * **(d)** How is the "Syntax-aligned composition" implemented? Since syntactic similarity is difficult to evaluate, it is not clear how the optimization is performed.
   * **(e)** It would be beneficial to provide a detailed instance in this section to illustrate what $y_2$ is, how to select a subsequence $y_2''$, and how to further enforce semantic preservation in that specific instance.

2. **SimPO Preference Probability (Critical)**
For SimPO, the preference probability is defined as $p(y_2 > y_1 | x) = \sigma(r(x, y_1) - r(x, y_2) - \gamma)$. Similarly, we expect $p(y_1 > y_2 | x) = \sigma(r(x, y_2) - r(x, y_1) - \gamma)$. Under this situation, the summation of probabilities does not equal 1. Does this imply a probability of a "tie"?

3. **Experimental Robustness (Strengthening)**
The experimental results are not convincing enough. According to the paper, the two responses only differ by one word while the others remain the same. While I understand the importance of keeping the same syntax structure to control for length, a more natural approach would be to leave several blanks and choose semantically close words for each. It seems too restrictive to only keep one blank.

---

> ### Author Response · Authors · 2026-03-17
> **Rebuttal**
>
> **1) Clarification of Section 3: Pure Semantic Preference**
>
> (a)	Thank you for pointing out that $T’’$ is not defined, we have revised it in our manuscript.
>
> (b)	Thank you for the question. "further enforce semantic preservation" means $m(y_2'')=m(y_2)=M$. We realize that this is somewhat misleading and we have revised our manuscript.
>
> For (a) and (b), We have revise as follow:
>
> we define the length-aligned response
>
> $y_2''=\Pi_I(y_2)=[T_1'',...,T_{n_1}''],$
>
> where, $T_i''$ are the tokens of $y_2''$, $i=1,\ldots,n_1$, such that $y_2''$ is a subsequence of $y_2$ that contains $T_k^2$ and $m(y_2'')=m(y_2)=M$. Then, the log-likelihood difference between $y_2''$ and $y_2$ can be interpreted as a length preference under fixed meaning.
>
> (c) and (d) Thank you for the question. We realize that the definition of similarity is unnecessarily complicated and we further simplify this paragraph. The revision of our manuscript is also provided as follow:
>
> Since the core semantic content of $y_2$ is encapsulated in the anchor token $T_{i_2}^2$, we treat $y_1$ as a syntactic template. We then construct the syntax-aligned response $y_2'$ by substituting the anchor token of $y_1$ with that of $y_2$. Formally, let $y_1 = [T_1^1, \dots, T_{i_1}^1, \dots, T_{n_1}^1]$. We define:
>
> $y_2' := [T_1^1, \dots, T_{i_2}^2, \dots, T_{n_1}^1],$
>
> under the assumption that $y_2'$ remains grammatically correct. By this construction, $y_2'$ inherits the length and syntactic skeleton of $y_1$ while adopting the key semantic component of $y_2$.
>
> (e) Thank you for the question. We would like to clarify that selecting an optimal subsequence is intended as a conceptual procedure to illustrate the theoretical framework, rather than an implementable algorithm. Specifically, the selection of such a subsequence depends dynamically on the specific content of $y_2$ and the target length constraints on a case-by-case basis, making a generalized algorithmic implementation non-trivial. We leave the development of a formal, automated algorithm for subsequence selection as a subject for future research.
>
> **2) SimPO Preference Probability**
>
> Thank you for the insightful question. In our opinion, the answer could be yes or no, although this specific probabilistic interpretation is not explicitly detailed in the original SimPO paper.
> In the SimPO framework, the preference probability is defined as:
>
> $p(y_w > y_l \mid x) = \sigma(r(x, y_w) - r(x, y_l) - \gamma)$
>
> The paper does not explicitly define the complementary probability $p(y_l > y_w \mid x)$, not like what you mentioned, which leads to two possible interpretations:
>
> 1. **Binary Outcome:** If we assume that $p(y_l \leq y_w \mid x) = \sigma(r(x, y_w) - r(x, y_l) + \gamma)$, then the probabilities sum to 1. In this case, there is no formal "tie" state; rather, the decision boundary is simply shifted to require a margin of $\gamma$ for $y_w$ to be preferred.
>
> 2. **Symmetric Margin (Tie Interpretation):** If we assume a symmetric definition where $p(y_l > y_w \mid x) = \sigma(r(x, y_l) - r(x, y_w) - \gamma)$, then as you noted, the sum of probabilities is less than 1:
>
> $$\sigma(r_w - r_l - \gamma) + \sigma(r_l - r_w - \gamma) < 1$$
>
> Under this situation, the "missing" probability mass $P_{\text{tie}} = 1 - [p(y_w > y_l) + p(y_l > y_w)]$ represents a region of indifference or a "tie" where neither response sufficiently outperforms the other by the required margin.
>
> **3) Experimental robustness: “only one blank / one word change is too restrictive”**
>
> Thank you for the insightful question. We agree that “single-slot substitution” is the strictest instantiation and can look artificial. Our intent was to guarantee that length and syntactic gaps are exactly controlled ($\Delta_{len}=0$, $\Delta_{syn}=0$), so that the residual gap can be attributed to the intended semantic substitution. This strictness is explicitly motivated by the decomposition and the practical difficulty of constructing intermediate responses while enforcing semantic equivalence and isolating syntax/length.
>
> Importantly, the theoretical setup already allows the semantic difference to be carried by short token sequences (anchor tokens), i.e., not inherently restricted to a single word. Likewise, even in the current definition we already say “single word (or phrase)”.

---

### Author Response · Authors · 2026-03-17
**General Response**

Dear Editors and Reviewers,

We thank all reviewers for their careful reading and constructive feedback. We have revised the paper substantially to improve clarity, positioning, and experimental transparency. First, we clarified the theoretical presentation in Section 3 by refining the definitions of the intermediate aligned responses, simplifying the discussion of syntax alignment, and explicitly framing the decomposition as a working hypothesis for controlled attribution rather than as a claim of strict independence among length, syntax, and semantics. In particular, we now define the length and syntax effects as residual gaps under progressively tighter alignment constraints, which better reflects the fact that these factors can be correlated in practice.

Second, we improved the paper’s positioning relative to prior work. We now state explicitly that the PMO objective is equivalent to Soft Preference Learning under a simple reparameterization, and we clarify that our contribution is not a fundamentally new loss, but rather a controlled semantic preference framework and an analysis of probability matching, reference attenuation, and collapse behavior in that setting. We believe this framing better highlights the paper’s actual novelty and scope.

Third, we strengthened the presentation of the experiments and their limitations. We now explicitly mark the reported training results as single-run, avoid overclaiming significance for small differences, and add uncertainty estimates over evaluation instances where feasible. We also clarified the role of synthetic soft labels: these targets are intentionally constructed for a controlled diagnostic setting where naturally occurring calibrated preference distributions are not readily available at scale. Our goal is not to teach arbitrary “semantic facts,” but to probe whether preference optimization methods preserve uncertainty or collapse toward overly sharp preferences once non-semantic confounds are removed.

Finally, we made the larger-scale evidence more explicit. The revised paper now clearly highlights results on 8B-scale models, showing that the main trends are not limited to 1B–1.5B backbones. Across these revisions, our aim was to make the paper more precise about its assumptions, more transparent about its empirical scope, and clearer about the central claim: controlled semantic preference pairs provide a useful lens for studying how different preference optimization objectives behave beyond aggregate win-rate metrics.

Best,
The authors

---

### Decision · Action_Editor_qtzX · 2026-04-20

**Recommendation:** Reject

**Audience:**

Yes

**Audience Explanation:**

Yes. The paper addresses preference optimization and alignment in large language models, and the attempt to separate semantic effects from length and syntactic confounds is conceptually interesting. I expect some readers in the RLHF/alignment community would be interested in this perspective.

**Claims And Evidence:**

No

**Claims Explanation:**

The paper studies how preference optimization methods behave in a controlled setting where length and syntactic confounds are reduced, so that one can better examine semantic preference matching and collapse behavior. However, the current submission does not provide sufficiently convincing evidence for its central claims in this setting, especially in their empirical and methodological developments. In particular, the proposed controlled setting relies on synthetic preference targets and a highly simplified construction, and the connection between this setup and meaningful semantic preference behavior in realistic alignment settings remains insufficiently validated. In addition, several of the reported comparisons are based on single training runs, and the margins between methods are often modest, which limits how strongly one can support the paper’s broader conclusions about the relative behavior of preference optimization methods. Overall, while the paper raises worthwhile questions, I do not think the present evidence is yet strong enough to support the claims at the level required for TMLR.

**Resubmission Of Major Revision:**

The authors may consider submitting a major revision at a later time.